# Associations Between Street-View Perceptions and Housing Prices: Subjective vs. Objective Measures Using Computer Vision and Machine Learning Techniques

**Xiang Xu ¹, Waishan Qiu ²,\*, Wenjing Li ³, Xun Liu ⁴, Ziye Zhang ⁵, Xiaojiang Li ⁶ and Dan Luo ⁷**

1    School of Architecture, South China University of Technology, Gguangzhou 510641, China; arcity001@mail.scut.edu.cn;
2    Department of City and Regional Planning, Cornell University, Ithaca, NY 14850, USA
3    Center for Spatial Information Science, The University of Tokyo, Chiba 277-8568, Japan; liwenjing@csis.u-tokyo.ac.jp
4    School of Architecture, University of Virginia, Charlottesville, VA 22904, USA; xl4xw@virginia.edu
5    Paul and Marcia Wythes Center on Contemporary China, Princeton University, Princeton, NJ 08544, USA; ziyez@princeton.edu
6    Geography and Urban Studies, College of Liberal Arts, Temple University, Philadelphia, PA 19122, USA; xiaojiang.li@temple.edu
7    School of Architecture, University of Queensland, St. Lucia QLD 4072, Australia; d.luo@uq.edu.au
\*    Correspondence: wq43@cornell.edu; Tel.: +1-6178524013

**Abstract:** This study investigated the extent to which subjectively and objectively measured street-level perceptions complement or conflict with each other in explaining property value. Street-scene perceptions can be subjectively assessed from self-reported survey questions, or objectively quantified from land use data or pixel ratios of physical features extracted from street-view imagery. Prior studies mainly relied on objective indicators to describe perceptions and found that a better street environment is associated with a price premium. While very few studies have addressed the impact of subjectively-assessed perceptions. We hypothesized that human perceptions have a subtle relationship to physical features that cannot be comprehensively captured with objective indicators. Subjective measures could be more effective to describe human perceptions, thus might explain more housing price variations. To test the hypothesis, we both subjectively and objectively measured six pairwise eye-level perceptions (i.e., Greenness, Walkability, Safety, Imageability, Enclosure, and Complexity). We then investigated their coherence and divergence for each perception respectively. Moreover, we revealed their similar or opposite effects in explaining house prices in Shanghai using the hedonic price model (HPM). Our intention was not to make causal statements. Instead, we set to address the coherent and conflicting effects of the two measures in explaining people's behaviors and preferences. Our method is high-throughput by extending classical urban design measurement protocols with current artificial intelligence (AI) frameworks for urban-scene understanding. First, we found the percentage increases in housing prices attributable to street-view perceptions were significant for both subjective and objective measures. While subjective scores explained more variance over objective scores. Second, the two measures exhibited opposite signs in explaining house prices for Greenness and Imageability perceptions. Our results indicated that objective measures which simply extract or recombine individual streetscape pixels cannot fully capture human perceptions. For perceptual qualities that were not familiar to the average person (e.g., Imageability), a subjective framework exhibits better performance. Conversely, for perceptions whose connotation are self-evident (e.g., Greenness), objective measures could outperform the subjective counterparts. This study demonstrates a more holistic understanding for street-scene perceptions and their relations to property values. It also sheds light on future studies where the coherence and divergence of the two measures could be further stressed.

**Keywords:** street view image; subjective and objective perceptions; housing prices; machine learning; computer vision

## 1. Introduction

### 1.1. Street Environment and Property Values

Urban street is a vital medium for urban residents to thrive—its perceptual quality has significant impacts on residents' behaviors and quality of life [1,2]. On the one hand, physical disorder visible in the street (e.g., broken windows, abandoned housings, graffiti, and decayed street lighting) correlates to crime, decreases residents' sense of safety, and consequently lowers residents' willingness to live there [3,4]. On the other hand, a well-designed and maintained street environment increases residents' physical activities, lessens their stress [5], and improves their health [6,7] in part due to the outdoor thermal microclimate [8–10] or perceived safety [1]. Streetscapes affect pedestrians' route choices [11], perceived thermal comfort and walking comfort [12] influenced by heat exposure [13]. Streetscapes also affect driving safety as a result of sun glare effects in urban roads [14].

Most importantly, the physical appearance of street environments such as greenness quality [15], as well as the derived perception (e.g., sense of place) [16] can be linked to neighborhood socioeconomic status [15] such as housing prices [17–21] and price appreciation [22]. The hedonic price model (HPM) has been widely applied to quantitatively reveal to what extent the street environment affects house prices along with other neighborhood, location, and structure characteristics [21,23–25]. Poor street views can directly and indirectly relate to property value variance [26]. As an example of the indirect relation, less vegetation coverage can lead to a less-comfortable thermal environment that is correlated to decreasing housing prices [27]. Therefore, a better understanding of perceptual qualities of the street can help cities to improve public health, safety, quality of life, and sustainability including economic and environmental resilience against climate change [19].

Along this line, street-view perceptions have be measured subjectively, objectively, or in combination [28–30]. On the one hand, subjective measures are self-reported perceptions from survey questionnaires or interviews [4,29–32]. However, their definitions were found to be inconsistent across studies, while their results were difficult to provide instructive policy implications. On the other hand, the objectively measured counterparts, either come from land use data in Geographical Information System (GIS) [33], or visual indices extracted from street-view imagery (SVI) [19,25,34,35]. Although objective measures could be translated into intervention strategies, they might fail to capture the subtle human perceptions which are a subjective sensory information process [28,31,36].

### 1.2. Hypothesis and Knowledge Gap

We hypothesize that human perceptions have a subtle relationship to physical features that cannot be comprehensively captured by recombining objective indicators measured from SVIs [35]. Instead, the subjectively assessed perceptions using visual surveys might exhibit a stronger association with housing prices. To test the hypothesis, we proposed measuring pairwise (i.e., subjective vs. objective measures) human perceptions. Their coherence and divergences, and associations with housing prices can be compared to reveal the effectiveness of these two converse measurements.

Notably, we must acknowledge that the intention is not to make causal inference between street perception and housing prices. The research design is to address the divergent correlations with housing prices between the two measurements. No causal statement can be made, although association is the necessary condition for causality, and the theory from architecture, urban planning, sociology, and economics seem to suggest such

a causal relationship. However, it could be in the reverse direction, or it could be mutual relationship, or a confounding variable that affects both house price and the street environment, such as a policy beautifying the district and investing in new urban infrastructures. Nevertheless, this points to future studies where the causal relationship between housing price and the street environment is a very important topic to work on. Due to data limitations, full panel data for both SVIs and housing price are difficult to acquire, thus our results and implications should be limited to correlations.

Along this line, existing studies have not simultaneously stressed subjectively and objectively measured streetscape perceptions sufficiently, nor in the discussion related to housing prices. There exist at least three knowledge gaps.

First, the extent to which subjective and objective perceptions of the street environment complement or overlap each other in influencing property values has not been stressed in the HPM literature. Prior studies that investigated the effects of both measures concentrated on walkability and health [28,29,35] or urban design [37]. However, they indicated poor agreements between the two measures [33]. One study showed that objective measures of the urban environment had more significant relationships with pedestrian behaviors [28], while another study indicated that the two measures were complementing each other [29].

Second, prior studies of street environments and house prices focused on objective measures and took physical features such as street greenery and sky view as perception indicators [18,19,21,25]. Few studies have addressed the impact of subjectively measured street perceptions (such as safety and imageability) on surrounding housing prices.

Third, even within objective measures, comprehensive and eye-level street perception have not been addressed enough. First, limited by the availability of large-scale urban perception data, prior objective measures mainly relied on GIS data. They often took the acreage of, distance to, or accessibility of the amenities (e.g., parks and plazas, green areas, and lakes) as indices, which did not capture eye-level perceptions. Very few eye-level features such as the greenery and sky have been investigated. Many other important visual elements such as street furniture, pedestrians, and commercial signs were not considered. Second, only individual impact was tested while the collective effect of these elements on house prices was ignored.

### 1.3. Contribution

Our contribution was fourfold. First, we enriched the literature in urban design measures with a scalable, automated, and high-throughput framework using online survey, open-source SVIs, and deep learning. The framework was efficient and accurate in measuring both subjective and objective eye-level streetscape perceptions. Second, we filled in the gap between the subjectively measured perceptual qualities and property values, where most prior studies merely focused on the impact of objective measures. Third, we quantified the associations between objective perceptions and housing prices using more comprehensive street features than prior studies, which relied on no more than three individual features. Fourth, we investigated the divergence and coherence between these two types of street perceptions. With a comprehensive investigation into the relationship between both subjective and objective streetscape perceptions and property values, this study provides a scientific basis for policy makers, planners, and real-estate developers to adequately address the economic value of street environments. It is also an applicable tool for formulating street design and maintenance policy and studying housing price characteristics.

## 2. Related Works

### 2.1. Conventional Street Environment Measures in HPM Studies

The most measured feature—street greenery—is often measured using indicators that were not human perception centered, such as the number of trees fronting the

property [38], the size of tree canopy [39], the percentage of ground cover [40], or the distance to large green areas [41]. While the literature in street greenery and housing price have been well developed, only several have explored the effects of other individual visual elements, for instance, street lighting [42], open space [17], and ground traffic [43]. However, traditional indicators such as tree canopy cover cannot fully incorporate the human eye-level perceived streetscape due to large variances in the field and direction of view [21,34,44]. Additionally, objectively measured visual elements alone failed to comprehensively represent residents' experiences on the street [37].

### 2.2. Measuring Objective Streetscape Features from SVI

More recently, with deep learning and publicly available geo-referenced street-view imagery (SVI), emerging studies started to apply semantic segmentation to extract the pixels of various physical features from SVIs as indices for objectively measured streetscape perceptions. On the one hand, SVIs are different from conventional GIS datasets, as they reflect a ground-level view of the street [45]. On the other hand, SVI data are easier to obtain, provide finer resolution with more details, and have wider data coverage (e.g., often publicly available at the city scale) over traditional methods such as on-site auditing [46–48]. Additionally, crowdsourcing, computer vision (CV), and machine learning (ML) technologies have also proven their accuracy and efficiency for large-scale application [4,49]. In particular, new studies within this regard have objectively detected curb ramps [50], measured eye-level greenery view index (GVI) [34], counted pedestrian numbers [51], and predicted sun glare [14]. This novel trend integrating big data and street-level perceptions has potential capacity to enable a more human-centric understanding of urban form, streetscape, micro-level environmental comforts, and societal sustainability at a larger scale [52].

Several studies emerged to measure how objective measures of the eye-level perceptual qualities of streetscapes affect resident's daily lives and consequently their willingness to pay and housing prices. A study used computer vision to quantify the street-visible greenery as GVI and estimated its positive economic benefits on property value in Beijing [21]. Ye et al. [25] found that GVI obtained the third-highest and positive regression coefficient in the housing price hedonic model in Shanghai. Fu et al. [1] extracted the view indices for tree, sky, and building from Baidu panoramas in Beijing and Shanghai, and found that tree and sky view significantly related to higher house prices. Chen et al. [18] revealed the non-linear relationship between house prices and GVI in Shanghai that higher GVI were associated with higher value properties.

Nevertheless, although these new studies were able to address the associations between the objective measures of streetscapes and housing prices, they only accounted for less than three individual physical features. While the impact of greenery and sky view has been increasingly addressed, many other important streetscape features affecting housing prices have never been tested.

Additionally, prior studies failed to incorporate subjectively measured perceptual qualities. Perception is a seemingly comprehensive and subjective process of attaining awareness of sensory information [37]. The street environment comprises various visual elements, and these features individually cannot represent human-scale perceptions. Therefore, subjectively measured perception is likely to vary and complement the effects of objective measures, such as the recombination of visual elements or the individual view indices. No research exists to quantify the relationship between the eye-level and subjective assessment of streetscape perceptions and property values within the current trend of utilizing deep learning and SVI frameworks.

### 2.3. Lack of Urban-Scale Perception Mapping

The number of studies on objective street measures have increased since 2006 [53] as objective measurement exhibits advantages in translating perception results directly into actionable design interventions. However, the divergence and coherence between

subjective and objective measurement protocols need to be examined comprehensively. On the one hand, few studies have sufficiently assessed important physical features that affect human-scale perceptions of the street environment that may influence property value. For instance, HPM studies contained little discussion of transparent facades on the ground floor [37], urban furniture [11], pedestrians [54], and commercial signs [55]. On the other hand, people's perceptions of streetscapes can be complex and are not reflected by individual physical features. Subjective perceptions may be complex or subtly related to physical features [37].

Subjective measures such as perceived assessments could serve as a tangible and firsthand counterpart to objective measures, helping clarify or even corroborate the meaning of the objective measures, and possibly justifying the value of using both types of measurements. While the impacts of objectively measured street environment perception on housing prices have been explored in a few studies in recent years [19,21,25], very few have considered the impact of subjectively measured streetscape perception (i.e., eye-level perceived qualities) on housing prices. The few studies incorporating human perceptions [16,22,56] were all built on MIT Place Pulse datasets [1,2].

This is a result of the scarcity of large-scale urban perception data. Most existing data on the appearance and perceptual qualities of urban environments rely on low-throughput surveys [57,58,59]. For example, Ewing et al. [60] have quantified five subjective urban design qualities (i.e., imageability, enclosure, human scale, transparency, and complexity) with a small sample size and a low-throughput method (see Table 1). They correlated expert ratings to the number of physical features that appeared in video clips, which required extensive human labor—a single video clip could take an hour. Moreover, the results of conventional methods such as visual collage, mail or phone surveys [53] were not reliable. Individual differences would make the evaluation inconsistent [58]. Therefore, conventional survey methods are expensive, low throughput, and coarse in spatial resolution [55]. Their conclusions are limited to the particular sample conditions [45,46], which weakens their generalizability.

### 2.4. Crowdsourcing Visual Survey and Deep Learning for Perception Mapping

More recently, participants can evaluate images using experts or crowdsourcing with online data collections, which have largely increased the data availability for built environment perceptions [1,2,36,55]. At the same time, crowdsourced studies are ideal sources of the training dataset required by ML and CV frameworks to build scene understanding algorithms [4]. In turn, the trained scene understanding algorithm is useful to create fine-grained urban perception maps across geographical regions. For instance, in Place Pulse 1.0, Salesses et al. [2] measured the perception of "safety", "class", and "uniqueness" with thousands of geo-tagged images. In Place Pulse 2.0, Dubey et al. [1] collects more than a hundred thousand images and 1.2 million pairwise comparisons from 81,630 online volunteers regarding six perceptual attributes: safe, lively, beautiful, wealthy, depressing, and boring.

On the one hand, built on Place Pulse, a cluster of studies trained deep learning frameworks to predict urban perceptions. For example, based on Place Pulse 1.0, Naik et al. [4] predicted the perceived safety of street scenes by extracting the generic image features (i.e., low-level features such as hue, saturation, lightness (HSL) histogram and edge detection) with a scene understanding algorithm; Rossetti et al. [61] extracted both low- and high-level (e.g., sky and tree) features as explanatory variables to predict the six perceptions. They implied that high-level features (e.g., view indices) increased not only the fit but also the interpretability of predictions. Based on the Place Pulse 2.0 dataset, Zhang et al. [55] used ML algorithms to predict the six perception scores from SVIs using high-level features (i.e., streetscape elements). They also identified the visual elements that may cause a place to be perceived as different perceptions [55]. Kang et al. incorporate the perceptions to improve predicting housing prices [16] and appreciation rate [22] in US cities. Although this cluster of emerging studies have investigated the role of subjective

perceptions on housing prices, they relied on secondary perception data (i.e., the Place Pulse dataset) which is not appropriate for urban landscapes in China.

Notably, within this trend, fewer studies examined the correlations between human perceptions measured from visual surveys and housing prices. Naik et al. [56] implemented an idea to investigate the co-volution of urban appearance, socioeconomic outcomes, and housing costs. However, their SVI and housing cost data only partially overlap, so they could not make causal statements. Kang et al. [22] modelled a housing appreciation rate, but it was also limited to single-year street view image. However, no study was able to reveal the causal relationship between housing prices and street perception using a panel dataset/time series dataset of both SVIs and housing prices, which points to a very important area for future studies.

On the other hand, a group of studies started to assemble their own perception data by recombing objective indicators extracted from SVIs. For example, Zhou et al. [62] constructed an Integrated Visual Walkability Index with four sub-indicators (psychological greenery, visual crowdedness, outdoor enclosure, and visual pavement) comprised of physical feature indices extracted from Baidu SVIs. Ma et al. [35] formed five objectively measured perceptions openness, greenness, enclosure, walkability, and imageability) to inform the effectiveness of urban renewal. Wang et al. [14] asked ten experts to subjectively score ten greenspace quality measures (e.g., accessibility, maintenance, variation, naturalness, colorfulness, safety, and general impression) of 2000 training images collected from Guangzhou, China and revealed greenspace exposure disparities are linked to neighborhood socioeconomic status including local hukou, education, unemployment, occupation, and housing condition. The above advancements using deep learning frameworks and SVI data for either subjective [2,4] or objective [35,55,62] streetscape perception measures were only concentrated on the urban design and walkability literature. We aim to fill the gap where no systematic investigation of the impacts of subjective and objective measures of streetscape perceptions on housing price has been conducted.

## 3. Data and Methods

### 3.1. Research Framework and Study Area

#### 3.1.1. Conceptual Framework

To what extent subjective and objective environmental measures are complementary or conflicting is never clearly stated in the housing price literature. Three clusters of emerging studies are particularly relevant to construct our framework: (1) the definitions of five subjective perceptual qualities [37,60], (2) the method to quantify five objectively measured perceptual scores [35], and how computer vision and machine learning are efficient to understand street scenes using street view images [4]. Based on these prior studies, six perceptual qualities (i.e., Greenness, Walkability, Safety, Imageability, Enclosure, and Complexity) were chosen. On the one hand, agreements of their significances in affecting residents' behavior, as well as their qualitative definitions have converged in the literature. On the other hand, their operational definitions for objective measures have also been achieved (see Table 1). Figure 1 illustrates the conceptual framework of the associations between human perception and housing prices. It also lists the key literature that inspired our study.

First, the presence of physical features such as sidewalk, tree canopy, building, and people affects residents' perceived street design qualities such as Safety and Imageability. In turn, these physical features, together with the perceived qualities, influence residents' overall behaviors including decisions to walk, to stay, and to live there, and consequently affect the housing prices. Notably, there were disagreements on whether sense of walkability, safety, and comfort belong to the perceptual qualities [63] or actually count for individual reactions [37]. Since our focus is the impact of perceptual qualities on house prices, we treat Safety and Walkability as perceptions (e.g., like Imageability) rather than

individual actions by Ewing and Handy [37]. This is consistent with Mehta [63] and Zhang et al. [55].

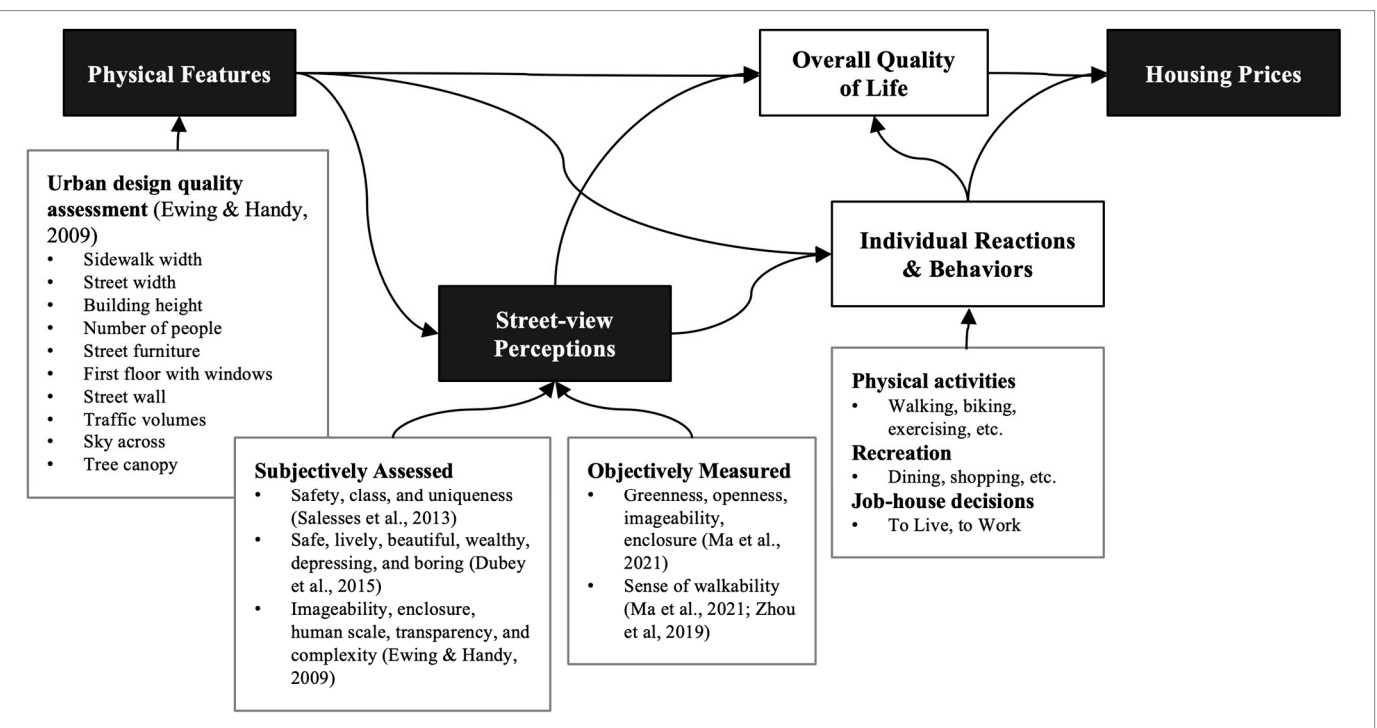

**Figure 1.** Conceptual framework. (Inspired by Ewing and Handy, 2009 [37]).

3.1.2. Analytical Framework

Subjective ratings of six perceived qualities were collected based on 300 randomly sampled SVIs across the Shanghai area (Figure 2). An online survey where a participant can choose her preferred street scene from pairwise SVIs regarding a perceptual quality question such as "Which place looks greener?" was carried out to extract the subjective ratings from 45 participants. Such a crowdsourced survey method has been proven to efficiently and accurately reveal people's true preferences [2]. Because the definitions of the six perceptions are not self-evident to the average person, it was not feasible to ask a random sample of residents to rate street environments with regard to qualities such as "imageability" [37]. Therefore, all participants were graduate students in Architecture, City Planning and Landscape Architecture who attended a design workshop [64]. They comprised an expert panel similar to Ewing and Handy [37]. The spatial location of SVIs was not revealed to participants and the pairwise comparison did not allow a draw. Second, the pairwise preferences were translated to ranked perception scores using the Microsoft TrueSkill [4,65] rating algorithm. TrueSkill is a Bayesian skill rating system [65,66] that provides balance between reliable ranking scores and size of participants. On average, every SVI in the survey only needed to be compared to the other 15 SVIs for the scores to converge. Third, we extracted and quantified the percentile indices of approximately thirty important physical features using a semantic parsing deep learning framework. We then trained ML models to predict the perceived scores using the physical feature indices extracted from the images as independent variables. The perception score prediction algorithms achieved high accuracy rates. Fourth, we predicted the six subjective perceptual scores for all SVIs sampled across the Shanghai metropolitan area (in total 25,276 images) based on their semantic feature pixel indices.

Last, we added both the subjective rating scores and objectively measured street feature indices to a hedonic price model with other important structural, locational, and neighborhood attributes using housing transaction data in Shanghai collected from

HomeLink (Lianjia.com), which is the largest real-estate website in China [19,25]. The predictability of subjective and objective measures of street perceptions on housing prices was compared based on the HPM approach. Specifically, we compared their achieved standardized coefficients to investigate three questions. (1) To what extent would subjectively-measured eye-level streetscape perceptions explain housing price variations? (2) Are subjective measures more effective than the objective counterparts? (3) What are the divergence and coherence between the two measurements?

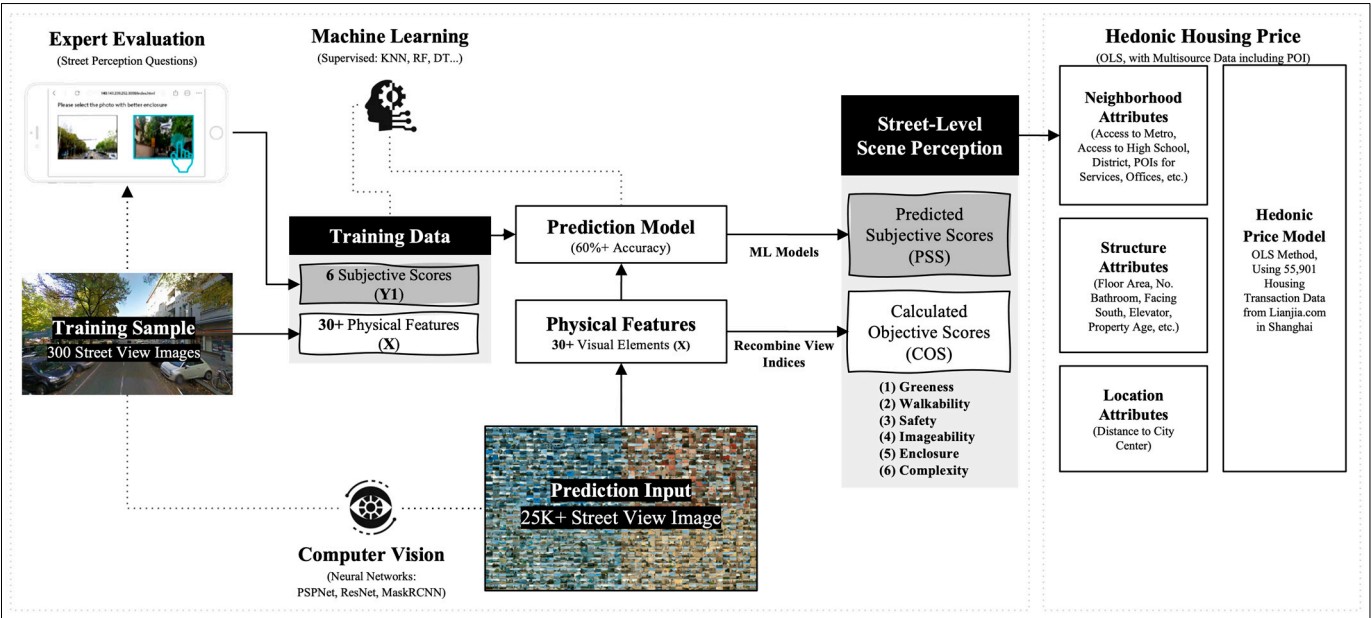

**Figure 2.** Analytical framework.

### 3.1.3. Study Area

As a financial hub of China, Shanghai's population density is 17,000 people/km². Since the housing reform in 1998, Shanghai has become one of the most costly and vibrant real-estate markets in China [18]. An empirical analysis for the city-wide area of Shanghai would provide essential implications for relevant studies.

### 3.2. *Selection of the Six Perceptual Qualities*

Prior studies have revealed that the level of walkability [67,68], greenery [21,25], openness [18,19], safety [4,69,70], aesthetics [26,71,72], and risks [70] in a neighborhood are all correlated with residents' daily behaviors, which eventually would affect the housing prices. Built on prior studies for walkability measurement [37] and urban renewal assessment [35] that integrated open source SVI data and deep learning frameworks, we selected six "operationalized" street-view perceptions: (1) Greenness, (2) Walkability, (3) Safety, (4) Imageability, (5) Enclosure, and (6) Complexity to present eye-level streetscape qualities. Table 1 lists their definitions and underlying physical feature determinants.

In urban design and urban scene understanding literature, both agreements and divergence exist regarding the qualitative definitions and quantitative methods for each perception. First, Ewing and Handy [37] provided the qualitative definitions for five of the six perceptions selected in this study (i.e., imageability, enclosure, transparency, human scale, and complexity). They statistically revealed the contributing streetscapes (like trees and pedestrian) that affect subjective perceptions collected from expert panels [37]. Their study laid the theoretical and operational foundations regarding measuring subjective perception for this study. Second, Ma et al. [35] objectively measured five perceptions (i.e., greenness, openness, walkability, enclosure, and imageability) by re-combining pixel indices of important streetscape elements extracted from SVIs using a deep learning

framework. Their study provides the foundation for objective perception measurement for this study. Third, prior studies also indicate that greenness [18,19,21,25] and walkability [67] are significantly related to property value, therefore we incorporated these two perceptions. Fourth, to avoid multicollinearity issue, perceived openness was not selected as it is opposite to perceived enclosure. They both focus on vertical elements in street view, and by operation sky view and building view are among the most significant elements. Last, safety (regarding violent crimes) has been consistently indicated to affects residents' behaviors and is associated with housing prices [3,4,55,63,73]. For instance, property value is highly discounted in districts perceived as insecure from surveys [73] or with higher crime rates [69]. Therefore, safety was included to complement the other five operationalized qualities to advance the measurement.

To quantify the subjectively perceived and objectively measured scores for these six perceptions, we took a twofold approach. On the one hand, subjective scores are predicted from SVIs using ML models, built upon the results generated by an online survey as training data. Notably, prior scene understanding deep learning frameworks [4,49,55] have largely inspired our approach. On the other hand, objective scores are calculated by integrating the indices of selected physical features extracted from SVIs [35,37,60,62]. Table 1 also includes the definitions and formulae to construct the six perceptions.

**Table 1.** Definitions and equations for six perceptual qualities.

| Perceptual Quality | Qualitative Definition | Significant Physical Features | Subjective Score Questions | Objective Score Equations (Based on Their Operational Definitions) |
|---|---|---|---|---|
| **1. Greenness** | Urban green spaces that are an essential element in streetscape, including forests, greenbelts, and lawns [35] | Tree view [34,35,45] | Which place looks greener? | The proportion of green space intermixed with building façades [35] $O1\_Greens_i = VI_{tree}$ (2.1) |
| **2. Walkability** | The psychological impact of the surrounding visual elements on the walking experience, such as the sense of comfort and pleasure for walking [35] | Pavement, sidewalk, fence, tree, grass [35,62] | Which place looks more Walkable? | The proportional relationship between the pavement, fence, and the overall road on walking experience [35] $O2\_Walkb_i = \frac{VI_{sidewlk}+VI_{fence}}{VI_{road}}$ (2.2) |
| **3. Safety** | An individual's experience of the risk of becoming a victim of crime and disturbance of public order [74] | Visual and physical connection and openness to adjacent spaces, physical condition and maintenance, lighting quality in space after dark, presence of surveillance cameras, security guards, guides, ushers, etc. | Which place looks safer? | Perceived safety from crime is affected by the physical condition and maintenance, the configuration of spaces, the types of land uses, the alterations and modifications made to the environment, and the presence or absence of, and the type of, people [63] $O3\_Safty_i = VI_{persn} + VI_{signb} + VI_{strlgt} + VI_{fence} + VI_{windwp}$ (2.3) |
| **4. Imageability** | The quality of a place that makes it distinct, recognizable and memorable [37]. | People, proportion of historic buildings, courtyards/plazas/parks, outdoor dining, buildings with non-rectangular silhouettes, noise level, major landscape features, buildings with identifiers [37] | Which place has better Imageability? | The proportions of the buildings, signs, and symbols as a proxy of street richness and diversity [35] $O4\_Imgbl_i = VI_{bldg} + VI_{skycrp} + VI_{signb}$ (2.4) |
| **5. Enclosure** | The degree to which streets and other public spaces are visually defined by buildings, walls, trees, and other vertical elements [37] | Proportion of street wall, proportion of sky, long sight lines, proportion of sky ahead [37] | Which place has better Enclosure? | The degree to which street canyons are visually enclosed by the sides of buildings, walls, trees and other vertical elements and the space of the horizontal ground between them [35] $O5\_Encls_i = \frac{VI_{bldg}+VI_{tree}}{VI_{road}+VI_{sidewlk}+VI_{earth}+VI_{grass}}$ (2.5) |
| **6. Complexity** | The visual richness of a place, which depends on the variety of the numbers and types of buildings, ornamentation, landscape elements, street | People, buildings, dominant building colors, accent colors, outdoor dining, public art [37] | Which place has better Complexity? | The numbers and kinds of buildings, architectural diversity and ornamentation, landscape elements, street furniture, signage, and human activity [60] $O6\_Cmplx_i = \frac{VI_{persn}+VI_{signb}+VI_{strlgh}+VI_{tree}+VI_{chair}+VI_{windwp}}{VI_{bldg}+VI_{road}}$ (2.6) |

|  |
|---|
| furniture, signage, and human activity [37] |

Notes: (1) $VI_{feature}$ denotes the view index of a physical feature (proportion of the visual element's pixels in a street-view imagery (SVI)), and is calculated as: $VI_{feature} = \frac{\sum_{i=1}^{n} Pixel_{feature}}{\sum_{i=1}^{m} Pixel_{total}} = \frac{1}{n}\sum_{i=1}^{n} Pixel_{feature}$, $feature \in \{tree, building, sky, etc\}$ [1]. (2) $VI_{tree}$, $VI_{sidewlk}$, $VI_{fence}$, $VI_{road}$, $VI_{persn}$, $VI_{signb}$, $VI_{strlgt}$, $VI_{windwp}$, $VI_{skycrp}$, $VI_{earth}$, $VI_{grass}$, $VI_{chair}$ denotes the view index of tree, sidewalk, fence, road, person, signboard, streetlight, windowpane, skyscraper, earth, grass, and chair, respectively.

### 3.3. Scoring Subjective and Objective Perceptions

SVIs provide a horizontal perspective of the street environment, which is closer to a pedestrian's eye-level perception [34,45], and therefore it becomes an ideal data source for the measures of human-centered streetscapes (Figure 3a). Prior studies have established efficient frameworks to predict subjective or objective perceptual scores from SVIs. On the one hand, Naik et al. [4] demonstrated that the combination of generic image features and the scores of perceived safety from a crowdsourced study can accurately predict the safety scores of streetscapes not used in the training dataset. Their methods for the subjective score prediction have significant inspiration for this study. On the other hand, many studies [35,37,62] objectively measure seemingly subjective urban design qualities such as enclosure, complexity, greenness, and walkability based on establishing the statistic relationships between crucial physical features and the quality ratings; they provide an operationalized framework for the objectively measured scores in this study.

Five steps were conducted to calculate both subjective and objective perceptual qualities from SVI: (1) downloaded SVIs from sampled sites; (2) collected and converted expert ratings to ranked scores as training labels using an online visual survey; (3) extracted pixel indices of different visual elements from SVIs as independent variables; (4) trained ML models to predict subjective scores; and (5) calculated objective scores based on formulae (Table 1) that recombine view indices of selected visual elements.

#### 3.3.1. Collection of SVIs

We have sampled SVI data at intervals of 50 m [35] along the centerline of public streets (i.e., outside gated communities and resident blocks) within a 1 km radius of a property's coordinates. On the one hand, only public street data are available. On the other hand, the green space inside gated communities is designed and developed by real-estate developers [75,76], a practice of land-speculation-oriented local entrepreneurialism [77]. Developers also compete to offer good landscapes and environments to lure buyers [78]. Excluding the "interior" street environment will alleviate the endogenous issue of housing prices on inner-community environmental design.

The typical block size in Shanghai is 6.8 hectares [79], with block width and length ranging between 300 and 500 m. Therefore, a 50 m interval will ensure 6–10 images sampled for each block edge. A 1 km radius was determined because Chinese cities advocate a 15 min walking distance for delineating a neighborhood which is approximately 1 km [62]. The sampling was processed in ArcGIS.

Each sample's SVI was downloaded by feeding coordinates into Baidu Street View API [80,81], which is the most used web map service in China. To ensure a consistent view angle, we maintained the same camera settings and image resolution (Figure 3b). In addition, for each SVI, we also selected similar a shooting time (summer and fall 2017) identified by filename to eliminate the seasonal variance in street environments. In total, we downloaded 25,276 valid SVIs.

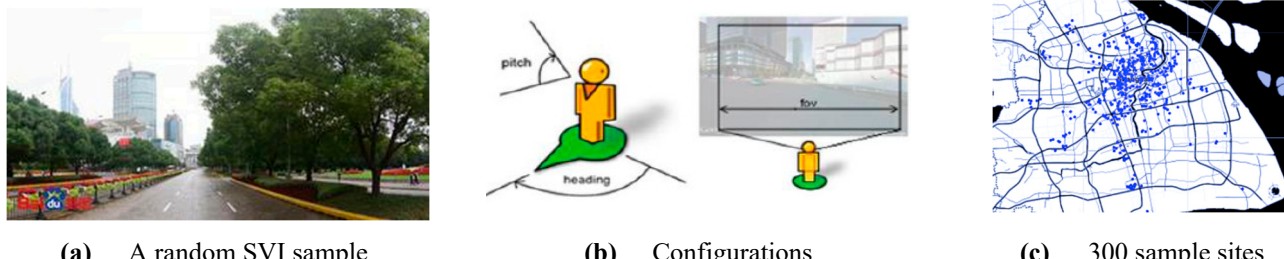

**(a)**　A random SVI sample　　　　　　**(b)**　Configurations　　　　　**(c)**　300 sample sites

**Figure 3.** Downloading SVIs from Baidu. (**a**) A SVI sample selected randomly. (**b**) Consistent viewpoints: the "heading" (view direction) was set parallel to the street centerline, the "FOV" (horizontal field of view) was 120 degrees, and the "pitch" (the up or down angle of the camera) was 0 degree. In addition, the resolution was 640 × 360 pixels. (**c**) 300 training images were sampled across a range of citywide locations in Shanghai.

3.3.2. Collecting Subjective Perception Scores from the Online Survey

To collect training labels reflecting subjective streetscape perceptions, we adopted a high-throughput urban scene understanding framework [2,82] that integrated crowdsourced survey data, deep learning, and ML. We built a survey website, where participants were shown two pairwise SVIs and were asked to click on the preferred image in response to six perceptual evaluation questions (Figure 4a). Taking Imageability as example, we first gave a qualitative definition on this quality. Participants were then asked, "Which place has higher degree of Imageability?"

To ensure the SVIs shown in the survey capture a variety of streetscapes from city center, suburban, to countryside, we randomly sampled 300 SVIs across Shanghai (Figure 3c). The 300 samples was chosen to balance between prediction accuracy and participants' workload. On the one hand, prior studies in statistics [83] revealed that the training sample size needed to train good models is at least 75 to 100 [84], or approximately ten times the number of parameters [85]. Given that we extracted over 30 streetscape elements from SVIs, the sample size of 300 would be sufficient. On the other hand, recall that in the survey, participants' pairwise preferences were converted to ranked scores with the TrueSkill algorithm [65,66]. On average, every SVI in the survey needed to be compared to 15 other SVIs for TrueSkill to converge [4]. Therefore, the 300 sample SVIs converged to stable ranked scores when we collected 4426 pairwise clicks by 45 participants in total. On average, each participant looked at approximately 100 pairwise photos, which was a reasonable workload for the individuals. Scores were then normalized to a 0–10 scale, where 1 is the best and 10 is the worst. These 300 SVIs labelled with six subjective scores became the training labels that we later used to train ML algorithms to predict subjective scores for all other unranked SVIs.

Regarding results of rater preferences, for perceived Greenness, participants preferred more greenery including trees, plants, and grass; for perceived Imageability, participants seemed to prefer streetscapes with iconic buildings and landmarks; for perceived Walkability and Enclosure, street views with less sky exposure, and more sidewalks, tree canopy and plants are preferred; for perceived Complexity and Safety, scenes with commercial activities on the ground floor are chosen (Figure 4b).

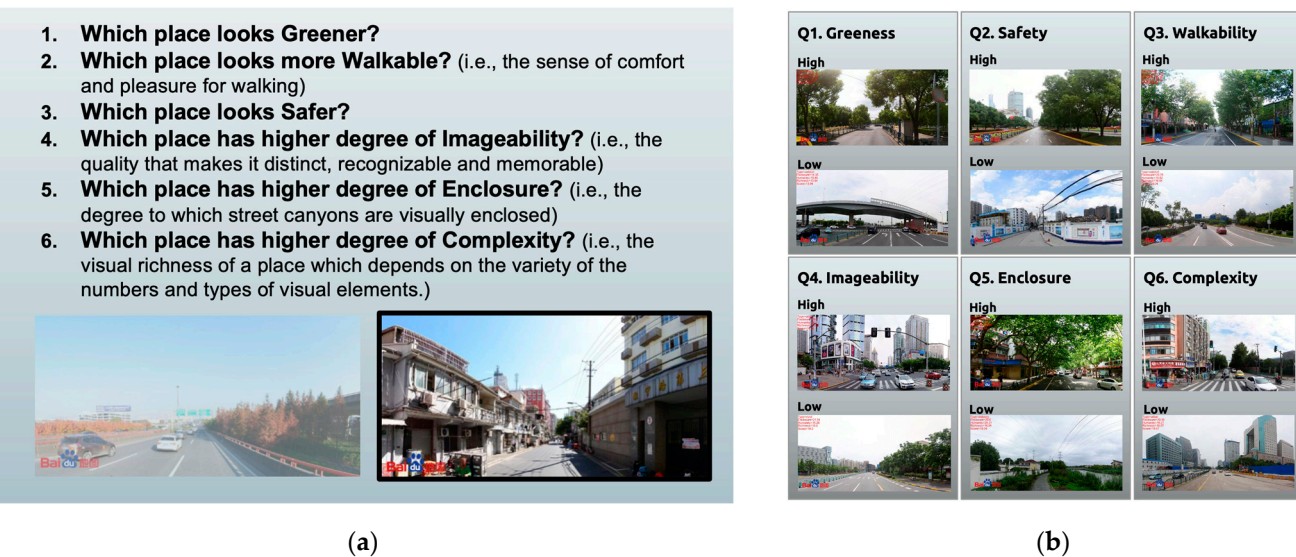

(**a**)  (**b**)

**Figure 4.** An online survey platform to collect subjective streetscape perceptions. (**a**) The website invited participants to choose one of pairwise images in response to the six evaluative questions. (**b**) Examples with a high and a low score for the six subjective perception qualities.

### 3.3.3. Classification of the Physical Features

Physical features, extracted from streetscape that lead to different perception qualities, have been statistically identified by previous studies [35,37,55,62]. Particularly, prior studies often utilized the view index of individual visual element as an important indicator, which was calculated by the percentage of the feature's pixels to the total pixels in an SVI [19]. For instance, building view index can be defined as the percentage of building pixels in an SVI. The importance of a visual element can be measured by view index through a pedestrian's eye-level view. Therefore, the appearance of various physical features in SVIs can be measured by the general formula (1) as follows.

$$VI_{feature} = \frac{\sum_{i=1}^{n} Pixel_{feature}}{\sum_{i=1}^{m} Pixel_{total}} = \frac{1}{n} \sum_{i=1}^{n} Pixel_{feature}, feature \in \{tree, building, sky, etc\} \tag{1}$$

where $VI_{feature}$ is the view index of a physical *feature*, $\sum_{i=1}^{n} PIXEL_{total}$ is the total number of pixels, and $\sum_{i=1}^{n} PIXEL_{obj}$ is the number of pixels related to the physical feature in an SVI.

To extract and calculate the view index of each feature from SVIs efficiently, a Pyramid Scene Parsing Network (PSPNet), which addressed object recognition and classification at a pixel level [86], was applied. Recently, the PSPNet framework achieved remarkable progress in semantic segmentation—it reached more than 93.4% pixel-level accuracy, and has been applied by multiple studies to extract features for property value [19]. Figure 5 showed SVIs randomly sampled with their results.

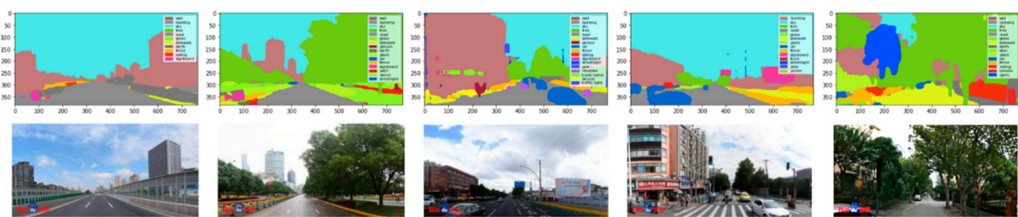

**Figure 5.** Five random sample results achieved by computer vision (CV) segmentation. Pairwise photos demonstrate semantic segmentation results and the raw SVIs.

### 3.3.4. Prediction of Subjective Perception Scores

With the 300 training images whose ranked scores from the online survey were taken as labels and the extracted view indices as explanatory variables, we trained and compared various ML models to predict the six subjective scores.

In terms of the selection of predictors, we notice the trend is to integrate the "low-level" (or generic) features [61,87,88] such as HSL histogram, saturation histogram, blob detection, and edge detection, with "high-level" features (i.e., the streetscape objects) [87–89]. Such a method has been proven to improve prediction accuracy. However, to align with the design practice [37] and provide more interpretability, we took a rule-based approach [55] which uses only high-level features for prediction.

The selection of features used in the final regression is first based on literature review in housing price studies and urban design studies with SVI and CV. Sky, tree, and building view indexes have been tested by prior housing studies [18,19,25], while other features such as person, sidewalk, car, fence were indicated by walkability studies [35,90]. Additionally, these ten objective streetscapes all have modest to large existence in street views, as well as large Gini importance scores.

The 300 samples were split to 80% and 20% for training and testing purposes, which was a common practice [55,84]. Widely applied ML models that have been proven to be efficient in predicting classes and continuous labels, especially those that have been tested in prior studies predicting streetscape perception scores using physical features extracted from SVIs, were tested [55], including Support Vector Machine (SVM), Random Forest (RF), Decision Tree (DT) and Gradient Boost (GB). To choose the optimal model, we compared model performances in terms of the R-square (R2) and the Mean Absolute Error (MAE). The performance of results were also compared to cross-study of similar methods and sample sizes in related fields [61,87,88].

### 3.3.5. Calculation of Objective Perceptual Scores

To construct objective perceptual scores, we followed the framework of Ma et al. [35] using equations to recombine important streetscape views (see Table 1). We took the percentile indices of those physical features including sky, tree, road, sidewalk views of each image to generate the objective scores for 300 training images and all the SVIs sampled across Shanghai. After calculation, these scores were normalized to a 0–1 scale (0 is the worst and 1 is the best) for interpretation purposes. While taking such an existing approach [35] without alterations, we must acknowledge that objective scores were constructed with arbitrary equations, which is a significant drawback. However, this is indeed our intention. We designed such a pairwise comparison to justify the hypothesis that "subjective measures of perceptions are more effective to represent users' sense of place and related behaviors", by comparing the pairwise subjective and objective measurement of the same concept.

### 3.3.6. Verification of Scores

For the 300 training images, their objective scores were also compared with subjective scores to investigate the coherence and divergence between the visually experienced perceptual qualities and the formula-derived qualities. Additionally, for both subjective and objective scores, Pearson's correlation analysis was also applied to validate the multicollinearity of the six qualities, respectively, to investigate the multicollinearity.

### *3.4. Hedonic Housing Price Model*

The HPM method assumes housing is a heterogeneous good whose price determinants can be investigated by regressing the house price on three main groups of explanatory variables capturing the property's structure, location, and neighborhood attributes [24]. The HPM method has been widely applied to quantify to what extent built environment factors affect property values [18,21,43]. Specifically, structure attributes are

comprised of variables illustrating the characteristics of the house including floor area, house orientation, building age, number of bathrooms, and elevator. Location characteristics are often characterized as the distance to the city center [91]. The accessibility of important urban facilities (such as trees, parks, plazas, metro stations, and health care, finance, and education services) is captured by neighborhood attributes. Although the scores for streetscape could be incorporated into neighborhood attributes, it is still necessary to divide them into a new group (STRE) to better reflect the effects of human perceptions. Therefore, the conventional HPM is extended as follows in our study:

$$\text{PRICE} = \alpha + \beta_1 \textbf{STRU} + \beta_2 \textbf{LOCA} + \beta_3 \textbf{NEIG} + \beta_4 \textbf{STRE} + \varepsilon \tag{2}$$

where PRICE is the housing price per square meter, $\beta_1$ to $\beta_4$ are the coefficients estimated for structure (**STRU**), location (**LOC**), neighborhood (**NEIG**), and streetscape (**STRE**) attributes, respectively, $\alpha$ is the constant, and $\varepsilon$ is the error term.

### 3.4.1. Housing Transactional Price

Transaction records of apartments occurring in 2019 within the municipality region of Shanghai were downloaded from Lianjia.com, a Chinese real-estate brokerage company which provide pre-owned apartment's information. The property's structure attributes and coordinates were included in the transaction records. The total 65,000 records were collected, and the dataset was cleaned for (1) outlier records whose transaction price seemed not trustworthy (e.g., zero value, or per unit price was more than ten times greater or smaller than the average); and (2) records lacking property attributes. In the end, 40,159 geo-tagged records were included for further studies, with an average price at 57,349 RMB/m$^2$. Figure 6a illustrated the price distribution. For the regression model, transaction price was then transformed to the natural logarithm form as the dependent variable [18,24].

### 3.4.2. Independent Variables

This study selected four categories of independent variables (Table 2) based on literature and data availability. Structure attributes included continuous variables such as number of bathrooms, building age, and total floor area. Categorical variables such as building height, building structure type, unit orientation, interior decoration quality, elevator were transformed to dummy variables.

With respect to location attributes, many studies have revealed that housing prices decreased as their distance to the city center increased [18,24,91,92]. Therefore, the road network distance from each property (1) to the central business district (CBD) of Shanghai and (2) to their nearest county center was calculated as locational attributes. Dummy variables were included to indicate the property's district or ring-road location to capture sub-market effects [92]. Through QGIS and Open Street Map (OSM), we calculated the distance based on 2018 road network data.

Density, distance to, and the accessibility of different urban amenities and services were captured in neighborhood attributes. POI density calculated the number of amenities such as retailing, restaurants, cafes, groceries, hospitals, and gyms per km$^2$ within the neighborhood's administration (district) boundary. Moreover, school district setting and education are influential factors on housing prices [93]: a good school district can bring a high price premium. Therefore, we incorporated 68 excellent educational facilities that are schools recognized by Shanghai government into calculation. Distance to the closest metro station and high schools was calculated by the road network. Accessibility was measured by counting the reachable numbers of metro stations or high schools within 1 and 5km [36], respectively. Neighborhood boundary was delineated based on Shapefile of Shanghai GIS in 2018. Data for public amenities and living services were extracted from Dazhongdianping.com in 2019 while metro stations and schools' data were from AutoNavi's map service in 2019, respectively (Figure 6b).

Regarding streetscape perception attributes, six ML-predicted subjective scores or the formula-derived objective counterparts were incorporated into HPM separately.

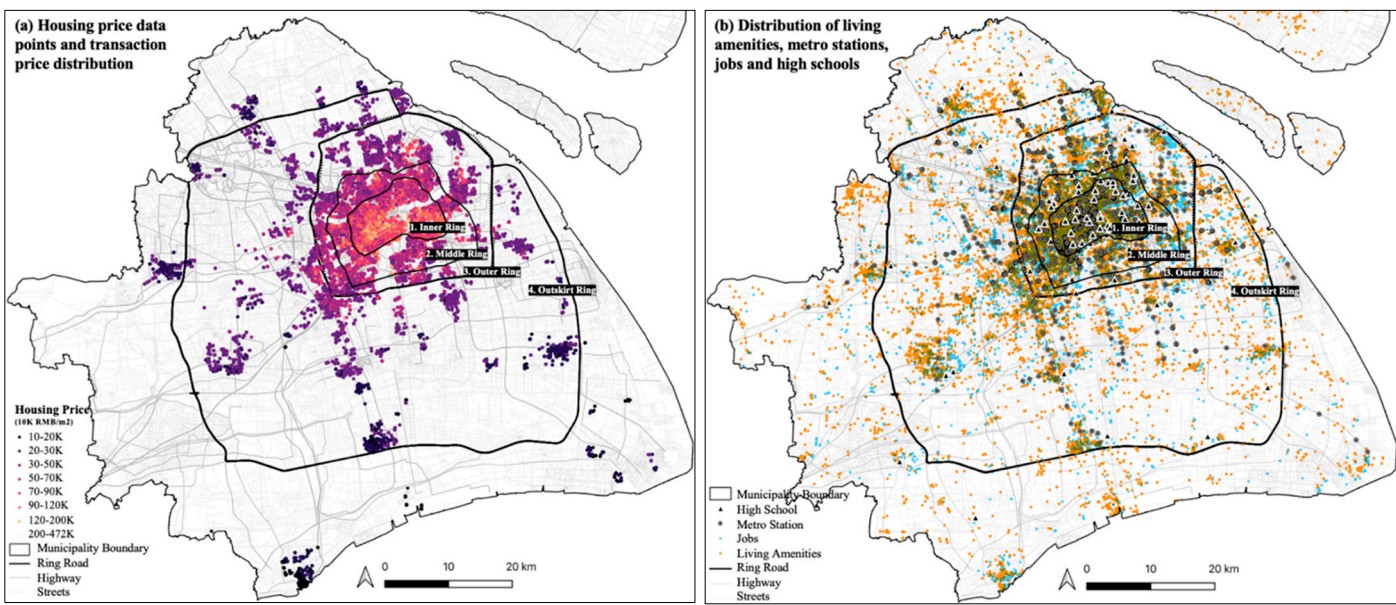

**Figure 6.** Spatial distribution of (**a**) 2019 transaction housing price data from HomeLink, and (**b**) neighborhood attributes including living amenities, metro station, job opportunities, and high schools.

**Table 2.** General descriptive statistics of the housing characteristics.

| Variables | Description | Count | Mean | Std. Dev. | Min | Max | Data Source |
|---|---|---|---|---|---|---|---|
| **PRICE** | Transactional price (RMB/m²) | 40159 | 57349 | 21683 | 10400 | 250813 | Lianjia.com |
| **STRUCTURAL ATTRIBUTES** | | | | | | | |
| **FLAREA** | Total floor area of the unit (m²) | 40159 | 85 | 43 | 15 | 588 | |
| **BEDRM** | Number of bedrooms | 40159 | 2.1 | 0.8 | 1 | 8 | |
| **LIVRM** | Number of living rooms | 40159 | 1.4 | 0.6 | 0 | 5 | |
| **KITCH** | Number of kitchens | 40159 | 1.0 | 0.2 | 0 | 5 | Web scraping from Lian-jia.com |
| **BATH** | Number of bathrooms | 40159 | 1.2 | 0.5 | 0 | 7 | |
| **TTLFLR** | Total floors of the building | 40159 | 11.0 | 7.9 | 1 | 62 | |
| **CSTRYR** | Construction year of the building | 40159 | 1998 | 9.4 | 1912 | 2019 | |

| | | Values | Count | % | Ave. Price (¥/m²) | Ave. Area (m²) | Data Source |
|---|---|---|---|---|---|---|---|
| **HGHT** | Categorial variables, on which floor in the building is the unit located? | Base | 1 | 0.0% | 34,452 | 87 | |
| | | High | 17,084 | 42.5% | 55,092 | 79 | |
| | | Low | 11,231 | 28.0% | 59,160 | 93 | |
| | | Mid | 11,843 | 29.5% | 58,891 | 86 | |
| **LAYT** | Categorial variables, the layout of the unit | Duplex | 1632 | 4.1% | 58,108 | 154 | |
| | | Flat | 38,527 | 95.9% | 57,317 | 82 | |
| **BTYPE** | Categorial variables, the size and shape of the building | Bungalow | 5 | 0.0% | 76,376 | 114 | |
| | | Mix | 207 | 0.5% | 72,013 | 106 | Web scraping from Lian-jia.com, converted to dummy variables with Python to dummies library |
| | | Slab | 36,379 | 90.6% | 56,346 | 85 | |
| | | Tower | 3568 | 8.9% | 66,706 | 88 | |
| **STH_NTH** | Categorial variables, is the unit south-facing? | Else | 7993 | 19.9% | 56,110 | 94 | |
| | | South | 32,166 | 80.1% | 57,657 | 83 | |
| **STRC** | Categorial variables, the structure of the building | Brick | 17,944 | 44.7% | 53,060 | 61 | |
| | | Other | 59 | 0.2% | 58,984 | 81 | |
| | | Steel | 22,156 | 55.2% | 60,819 | 105 | |
| **DÉCOR** | Categorical variable, the interior quality of the unit | Blank | 1903 | 4.7% | 47,779 | 84 | |
| | | Other | 2863 | 7.1% | 53,395 | 79 | |
| | | Refined | 20,859 | 51.9% | 61,322 | 96 | |

| | | | | | | | |
|---|---|---|---|---|---|---|---|
| | | Simple | 14,534 | 36.2% | 53,680 | 72 | |
| **ELEVTR** | Categorical variable, is an elevator availa- | No | 24,106 | 60.0% | 52,764 | 69 | |
| | ble? | Yes | 16,053 | 40.0% | 64,235 | 110 | |

**LOCATION ATTRIBUTES**

| | | Count | Mean | Std. Dev. | Min | Max | Data Source |
|---|---|---|---|---|---|---|---|
| **D2SCBD** | Network distance to its district center | 40,159 | 4.77 | 3.04 | 0.02 | 16.29 | Computed in ArcGIS, with |
| **D2CBD** | Network distance to the center (Bund) | 40,159 | 12.62 | 7.48 | 0.03 | 35.11 | Shanghai (2018) shapefile |

| | | Values | Count | % | Ave. Price (¥/m²) | Ave. Area (m²) | |
|---|---|---|---|---|---|---|---|
| **RING_X** | Categorical variable, within which ring road is the unit located? | Ring1 | 9290 | 23.1% | 81,151 | 88 | |
| | | Ring2 | 9835 | 24.5% | 63,057 | 79 | |
| | | Ring3 | 8742 | 21.8% | 52,356 | 81 | |
| | | Ring4 | 12,292 | 30.6% | 38,345 | 92 | |
| **CTY_XX** | Categorical variable, in which district is the unit located? The letters XX after CTY_ stands for the district name | BS: Baoshan | 3390 | 8.4% | 44,159 | 81 | |
| | | CN: Changning | 2400 | 6.0% | 70,051 | 83 | |
| | | FX: Fengxian | 992 | 2.5% | 24,524 | 95 | |
| | | HK: Hongkou | 1513 | 3.8% | 66,210 | 80 | |
| | | HP: Huangpu | 1267 | 3.2% | 92,725 | 103 | Web scraping from Lian- |
| | | JA: Jin'an | 964 | 2.4% | 95,101 | 90 | jia.com, converted to |
| | | JD: Jiading | 1662 | 4.1% | 37,527 | 87 | dummy variables with Py- |
| | | MH: Minhang | 4806 | 12.0% | 49,479 | 91 | thon to dummies library |
| | | PD: Pudong | 9389 | 23.4% | 57,590 | 87 | |
| | | PT: Putuo | 2941 | 7.3% | 58,412 | 76 | |
| | | QP: Qingpu | 678 | 1.7% | 30,976 | 94 | |
| | | JS: Jinshan | 2201 | 5.5% | 36,432 | 100 | |
| | | XH: Xuhui | 3060 | 7.6% | 74,879 | 79 | |
| | | YP: Yangpu | 3091 | 7.7% | 62,677 | 72 | |
| | | ZB: Zhabei | 1805 | 4.5% | 63,647 | 79 | |

**NEIGHBORHOOD ATTRIBUTES**

| | | Count | Mean | Std. Dev. | Min | Max | Data Source |
|---|---|---|---|---|---|---|---|
| **DENSRV** | Density of Living Service (thousand/km²) | 40,159 | 0.115 | 0.187 | 0 | 3.5 | from Dazhongdi- |
| **DENWRK** | Density of Office (thousand/km²) | 40,159 | 9.5 | 22.4 | 0 | 573.5 | anping.com, density calcu- lated in ArcGIS |
| **D2MTR** | Distance to Metro (km) | 40,159 | 0.8 | 0.7 | 0.01 | 7.8 | location data scraped from |
| **A2MTR** | Accessibility to Metro | 40,159 | 5.7 | 6.8 | 0 | 46.0 | GaodeMap.com, distances |
| **D2SCH** | Distance to School (km) | 40,159 | 2.7 | 2.3 | 0.02 | 11.9 | calculated in Python |
| **A2SCH** | Accessibility to School | 40,159 | 7.0 | 7.0 | 0 | 29.0 | |

**SUBJECTIVE STREETSCAPE ATTRIBUTES**

| | | Count | Mean | Std. Dev. | Min | Max | |
|---|---|---|---|---|---|---|---|
| **S1_GREEN** | Subjectively perceived greenness | 40,159 | 0.8 | 0.0 | 0.4 | 0.9 | |
| **S2_WLKBL** | Subjectively perceived walkability | 40,159 | 0.6 | 0.1 | 0.4 | 0.8 | Predicted with ML models |
| **S3_SAFTY** | Subjectively perceived safety | 40,159 | 0.7 | 0.1 | 0.3 | 1.0 | with physical feature view |
| **S4_IMBLT** | Subjectively perceived imageability | 40,159 | 0.7 | 0.1 | 0.3 | 0.9 | indices as independent |
| **S5_ENCLS** | Subjectively perceived enclosure | 40,159 | 0.7 | 0.1 | 0.3 | 0.9 | variables extracted from |
| **S6_CMPLX** | Subjectively perceived complexity | 40,159 | 0.6 | 0.0 | 0.5 | 0.9 | Baidu SVIs |

**SUBJECTIVE STREETSCAPE ATTRIBUTES**

| | | Count | Mean | Std. Dev. | Min | Max | |
|---|---|---|---|---|---|---|---|
| **O1_GREEN** | Objectively derived greenness | 40,159 | 0.4 | 0.1 | 0.0 | 0.8 | |
| **O2_WALKB** | Objectively derived walkability | 40,159 | 0.6 | 0.1 | 0.2 | 0.7 | Equation derived scores |
| **O3_SAFTY** | Objectively derived safety | 40,159 | 0.4 | 0.1 | 0.1 | 0.7 | by recombining selected |
| **O4_IMBLT** | Objectively derived imageability | 40,159 | 0.6 | 0.1 | 0.0 | 0.9 | physical feature view indi- |
| **O5_ENCLS** | Objectively derived enclosure | 40,159 | 0.6 | 0.0 | 0.1 | 0.7 | ces |
| **O6_CMPLX** | Objectively derived complexity | 40,159 | 0.3 | 0.1 | 0.0 | 0.6 | |

### 3.4.3. Model Architecture

The hedonic modeling comprised four steps. First, as a preliminary test, we added each group of attributes, namely, the (1) structural, (2) locational, (3) neighborhood, (4) subjective streetscape scores, and (5) objective streetscape scores into separate OLS models to understand the individual and collaborative contributions of each attribute group. Second, we constructed a base model using former three groups of attributes. No streetscape variables were included, and all insignificant variables were removed. Thus, we constructed the base model (Model 1). Third, based on Model 1, we added all six subjective scores (Model 2) and all six objective scores (Model 3) separately to examine the different

impacts between perceived scores and objectively derived scores. Durbin–Watson results were checked to ensure final models where autocorrelation effects were not significant. Variance Inflation Factor (VIF) was calculated to examine variables with correlation problems (VIF value > 10), of which less important variables with multicollinearity were removed [94]. The global importance of individual variables was tested with the Scikit-learn library in Python.

## 4. Analysis Results

### 4.1. Descriptive Statistics of the Segmentation

We calculated the view indices of more than thirty physical features from the 300 training images through a PSPNet pre-trained semantic segmentation algorithm according to the general formula (1). These view indices were regarded as the explanatory variables to predict six subjectively perceived scores (see Section 4.2.1) as well as inputs for the equation-derived objective scores (see Table 1 and Section 4.2.2). Prior urban design literature [35,37,55,62] has revealed that at least ten elements (building, sky, tree, curbs, roads, street wall, proportion windows, street furniture's, street lights, and signboard) were conceived to have significant effects on human perception among features shown in Table 3.

**Table 3.** Summary of the physical features extracted from the training street view images.

| Sort | Feature | Mean Value | Std. Dev. |
|------|---------|-----------|-----------|
| 1 | Sky | 39.68% | 17.11% |
| 2 | Tree | 21.75% | 17.66% |
| 3 | Road | 11.60% | 6.37% |
| 4 | Building | 11.52% | 13.83% |
| 5 | Plant | 2.15% | 3.86% |
| 6 | Wall | 2.06% | 5.37% |
| 7 | Sidewalk | 1.84% | 2.62% |
| 8 | Fence | 1.66% | 2.80% |
| 9 | Grass | 1.53% | 2.79% |
| 10 | Car | 1.52% | 2.58% |
| 11 | Earth | 1.11% | 2.84% |
| 12 | Ceiling | 0.61% | 5.09% |
| 13 | Railing | 0.35% | 1.31% |
| 14 | Bridge | 0.34% | 2.59% |
| 15 | Signboard | 0.26% | 0.88% |
| 16 | Water | 0.26% | 1.43% |
| 17 | Van | 0.09% | 0.67% |
| 18 | Person | 0.08% | 0.27% |
| 19 | Skyscraper | 0.08% | 0.78% |
| 20 | Streetlight | 0.06% | 0.16% |
| 21 | Column | 0.06% | 0.51% |
| 22 | Minibike | 0.05% | 0.29% |
| 23 | Bicycle | 0.04% | 0.26% |
| 24 | Awning | 0.02% | 0.30% |
| 25 | Ashcan | 0.01% | 0.09% |
| 26 | Windowpane | 0.01% | 0.32% |
| 27 | Mountain | 0.01% | 0.19% |
| 28 | Fountain | 0.00% | 0.14% |
| 29 | Pier | 0.00% | 0.08% |
| 30 | Chair | 0.00% | 0.04% |

| 31 | Booth | 0.00% | 0.05% |
|----|-------|-------|-------|
| 32 | Sculpture | 0.00% | 0.04% |
| 33 | Bulletin board | 0.00% | 0.06% |
| 34 | Lamp | 0.00% | 0.00% |
| 35 | Sofa | 0.00% | 0.00% |

*4.2. Subjective and Objective Scores and Correlation Analysis*

4.2.1. Subjective Scores

The performance of different ML models varied across six perceptual scores. GB outperformed other ML models in Greenness, Walkability, and Imageability scores, while SVM was selected to predict Enclosure and Complexity scores and RF performed best in predicting Safety scores (Table 4). The accuracies of the different subjective scores predicted varied. The accuracy rates for predicting Imageability, Greenness, and Complexity were slightly higher than that of the remaining three, which might be caused by how participants vary in different scene perceptions [55]. Participants tend to exhibit more similarity in what kind of street view is greener/more imageable/more complex. Another reason might be due to the small sample size (i.e., only 300 images rated by 45 participants). This points to an important area to be improved for future studies: (1) incorporating low-level features such as HSL, saturation, blob, and edge detection to complement high-level features can improve prediction accuracy [61,87,88]; (2) while collecting large training dataset with inputs of more raters can also improve the results.

Nevertheless, the prediction accuracy was acceptable. First, the range of R2s is between 0.47 and 0.51, meaning that all selected models explained approximately half of the variance, indicating a significant improvement from Ewing and Handy [37,60] and Park et al. [90], where the variance explained ranged from 0.21 to 0.37.

Second, MAEs ranged between 1.2 and 1.51, indicating the prediction errors would not offset fitted value far from true scores in the 0–10 range. In other words, if we transform the 0–10 scale scores to categorical labels, e.g., 0–2.5 terrible, 2.5–5 normal, 5–7.5 good, and 7.5–10 very good, then the interval of each category is 2.5: apparently, predictions of MAEs in the range 1.2–1.51 are within the categorical interval, and the accuracy of predicting correct labels would be acceptable. Notably, our MAE performance is relatively better than that of Yao et al. [89] (MAE $\in [1.597 - 3.282]$).

Third, we tried to justify our results with cross-study comparison to relevant studies with similar/comparable sample size (in terms of numbers of raters and training sample) and modeling method (i.e., integrating ML/deep learning framework with visual SVI surveys). Regarding R2, the performance of our models is close to that of Naik et al. [82] (average R2=0.568), partially better than Yao et al. [89] (R2 $\in [0.34 - 0.76]$), and stronger than that of Ito and Biljecki [88] (R2 below zero) who also locally collected own perception training samples, and is relatively lower than Dubey et al. [1] and Verma et al. [87] (R2=0.56-0.79).

**Table 4.** Performance of machine learning algorithms.

| | S1_Green | | S2_Wlkbl | | S3_Safty | | S4_Imblt | | S5_Encls | | S6_Cmplx | |
|---|---|---|---|---|---|---|---|---|---|---|---|---|
| Model | $R^2$ | MAE (std) | $R^2$ | MAE (std) | $R^2$ | MAE (std) | $R^2$ | MAE (std) | $R^2$ | MAE (std) | $R^2$ | MAE (std) |
| SVM | 0.39 | 1.46 | 0.51 | 1.35 | 0.41 | 1.25 | 0.24 | 1.79 | 0.48* | 1.51(0.6) | 0.49 * | 1.50(0.8) |
| Random Forest | 0.41 | 1.43 | 0.46 | 1.36 | 0.47* | 1.19 (0.7) | 0.29 | 1.73 | 0.43 | 1.55 | 0.27 | 1.63 |
| Decision Tree | 0.12 | 1.96 | 0.13 | 1.94 | 0.18 | 1.58 | 0.05 | 2.36 | 0.26 | 2.29 | 0.08 | 2.14 |
| Gradient Boosting | 0.49 * | 1.39 (0.6) | 0.48 * | 1.33 (0.7) | 0.47 | 1.21 | 0.51 * | 1.62(1.0) | 0.41 | 1.52 | 0.14 | 2.01 |

Note：* denotes the best-performing model selected to predict scores; (#) reports the std.dev. of the best model prediction.

The best-performing models were applied to predict the six subjective scores for the 25,276 SVIs, respectively. Because the objective counterparts were derived with view indices that ranged from 0 to 1, to make these two sets of results comparable, subjective

scores were also re-scaled to the 0–1 range. We then assigned the predicted scores to the property data points by taking the average scores from the SVIs located within the 1 km radius of the property to represent the average quality of a 15 min walking distance which describes surrounding neighborhood [62].

In addition, both urban design theory [37] and statistic inference suggest that not all visual elements extracted from SVIs are relevant to predicting perceptions. Therefore, we ranked the global importance (GI) of individual elements using Tree-Based Regressor with Scikit-learn in Python [4,18]. GI computed how much each variable contributes to decreasing the weighted impurity, thus providing the importance score. As a result, view indices of sky, tree, building, car, and road ranked highest in their sum importance (Table 5), which is consistent with prior findings in urban design [37]. Figure 7a reported the top 15 important features and their GI score for each subjective perception, respectively. Surprisingly, several visual elements that have been proven to be important, such as person, sidewalk, signboard, and street furniture, were not in the top ten for predicting Walkability (Figure 7b).

**Table 5.** Features global importance (GI) in predicting six subjective scores.

| Feature | S1_Green Imp. Score | Sort | S2_Wlkbl Imp. Score | Sort | S3_Safty Imp. Score | Sort | S4_Imblt Imp. Score | Sort | S5_Encls Imp. Score | Sort | S6_Cmplx Imp. Score | Sort | Sum Importance Sum Score | Sort |
|---|---|---|---|---|---|---|---|---|---|---|---|---|---|---|
| sky | 0.033 | 8 | 0.183 | 1 | 0.197 | 1 | 0.162 | 1 | 0.492 | 1 | 0.139 | 1 | 1.205 | 1 |
| tree | 0.288 | 1 | 0.042 | 7 | 0.186 | 2 | 0.130 | 2 | 0.042 | 4 | 0.042 | 7 | 0.730 | 2 |
| building | 0.133 | 2 | 0.102 | 3 | 0.108 | 3 | 0.053 | 5 | 0.098 | 2 | 0.099 | 2 | 0.594 | 3 |
| car | 0.057 | 4 | 0.133 | 2 | 0.072 | 4 | 0.038 | 9 | 0.027 | 6 | 0.098 | 3 | 0.423 | 4 |
| road | 0.072 | 3 | 0.037 | 8 | 0.059 | 5 | 0.049 | 6 | 0.046 | 3 | 0.038 | 12 | 0.301 | 5 |
| wall | 0.032 | 9 | 0.030 | 10 | 0.041 | 7 | 0.066 | 4 | 0.021 | 9 | 0.054 | 4 | 0.244 | 6 |
| plant | 0.056 | 5 | 0.050 | 4 | 0.024 | 12 | 0.031 | 10 | 0.033 | 5 | 0.042 | 9 | 0.236 | 7 |
| grass | 0.044 | 7 | 0.029 | 11 | 0.015 | 13 | 0.073 | 3 | 0.022 | 8 | 0.044 | 6 | 0.228 | 8 |
| fence | 0.021 | 13 | 0.050 | 5 | 0.033 | 9 | 0.041 | 7 | 0.015 | 12 | 0.042 | 8 | 0.202 | 9 |
| earth | 0.048 | 6 | 0.048 | 6 | 0.024 | 11 | 0.031 | 11 | 0.017 | 10 | 0.027 | 13 | 0.196 | 10 |
| person | 0.026 | 10 | 0.028 | 13 | 0.036 | 8 | 0.040 | 8 | 0.022 | 7 | 0.038 | 11 | 0.191 | 11 |
| sidewalk | 0.025 | 12 | 0.026 | 15 | 0.050 | 6 | 0.029 | 14 | 0.016 | 11 | 0.042 | 10 | 0.188 | 12 |
| signboard | 0.018 | 14 | 0.034 | 9 | 0.030 | 10 | 0.024 | 16 | 0.015 | 13 | 0.027 | 14 | 0.147 | 13 |
| truck | 0.026 | 11 | 0.017 | 18 | 0.010 | 16 | 0.030 | 12 | 0.013 | 15 | 0.020 | 17 | 0.116 | 14 |
| bicycle | 0.010 | 18 | 0.025 | 16 | 0.005 | 21 | 0.013 | 20 | 0.006 | 23 | 0.046 | 5 | 0.104 | 15 |
| streetlight | 0.016 | 16 | 0.028 | 14 | 0.014 | 15 | 0.016 | 18 | 0.015 | 14 | 0.016 | 19 | 0.104 | 16 |
| railing | 0.017 | 15 | 0.028 | 12 | 0.015 | 14 | 0.010 | 21 | 0.011 | 16 | 0.020 | 18 | 0.102 | 17 |
| chair | 0.010 | 19 | 0.017 | 17 | 0.002 | 25 | 0.030 | 13 | 0.008 | 19 | 0.024 | 15 | 0.091 | 18 |
| minibike | 0.005 | 22 | 0.010 | 21 | 0.005 | 22 | 0.024 | 15 | 0.009 | 18 | 0.021 | 16 | 0.073 | 19 |
| mountain | 0.003 | 23 | 0.015 | 19 | 0.007 | 18 | 0.014 | 19 | 0.010 | 17 | 0.007 | 23 | 0.054 | 20 |

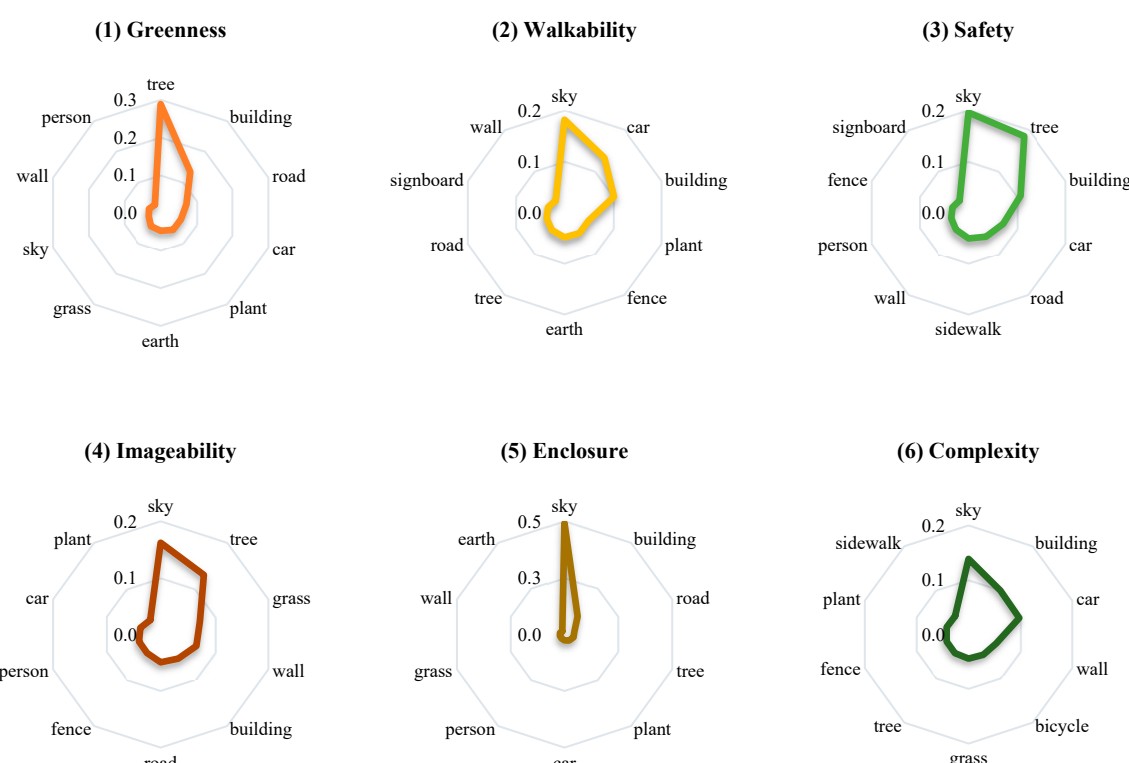

**Figure 7.** Important features in predicting six subjective scores. (**a**) Top 15 physical features for each subjective score and their sum importance. (**b**) Top 10 important visual elements in predicting each subjective score, respectively.

### 4.2.2. Objective Scores

Based on formulae defined in Table 1, six objective scores were generated by recombining the view indices of selected physical features. According to the formulae, there are significant differences in the quantity and proportion of dominant visual elements from different locations in Shanghai. Particularly, six objective scores have been affected by

variations resulted from physical features with vast and ubiquitous existence, such as trees, sky, buildings, and roads.

### 4.2.3. Correlation Analysis for Subjective and Objective Scores Respectively

Human perceptions can be complex and intertwined; therefore, intuitively, various perceptual qualities could be correlated. A previous study has shown that some pairs of perceptions measured from SVIs exhibited high correlation, including "beautiful–wealthy" and "depressing–safe" [55]. Therefore, we conducted Pearson's correlation analysis to validate the multicollinearity (Figure 8).

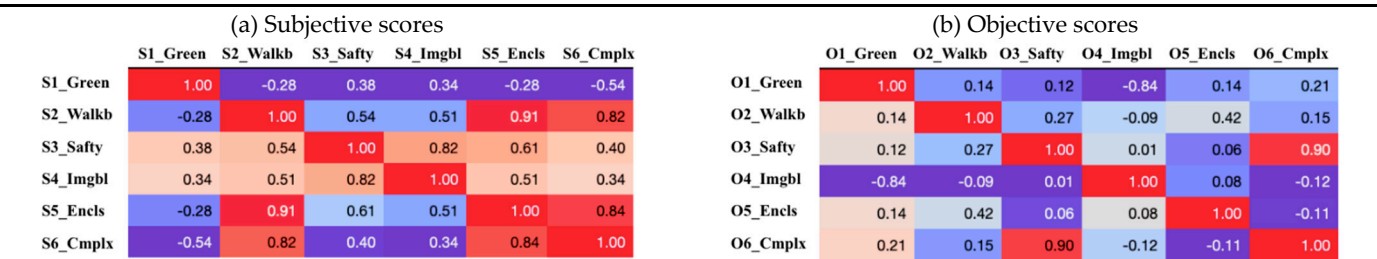

Note: All coefficients are significant at the 0.01 level.
To interpret coefficients: 0.0 to ±0.3: negligible; ±0.3 to ±0.5: low; ±0.5 to ±0.7: moderate; ±0.7 to ±0.9: high; and ±0.9 to ±1.0: very high.

**Figure 8.** Pearson's correlation coefficients among the (**a**) six subjective scores and (**b**) six objective scores.

On the one hand, several pairwise coefficients between subjective perceptual qualities, including Walkability–Enclosure, Walkability–Complexity, Safety–Imageability, and Enclosure–Complexity, have a moderate (between ±0.30 and ±0.5) to high (between ±0.50 and ±1) degree of correlation. This is consistent with [37] that enclosure and complexity was positively correlated to walkability. This is reasonable as they share similar qualitative definitions determined by common visual features such as sky, signboards, street furniture and persons. Conversely, objective scores mostly exhibit a low (between 0.1 and 0.3) to moderate degree of correlation. In other words, the formula-derived objective perceptions reduced multicollinearity. Notably, only the Safety–Complexity pair has a high degree of correlation. Meanwhile, complexity score should be excluded from hedonic price regression in later sections: it was highly correlated to at least one other score for both subjective and objective frameworks.

### 4.2.4. Coherence and Divergence between Subjective and Objective Scores

First, for both predicted and formula-derived scores, data were closed to normal distribution (Figure 9), indicating the perception qualities fit the most common and natural phenomenon in probability distribution. Second, three qualities, namely, Walkability, Imageability, and Enclosure, see more coherence in their mean value and variance. Third, the other three qualities, i.e., Greenness, Safety, and Complexity, exhibit more divergence in score distribution. People tend to overstate the perceived qualities since subjective score means are all significantly larger than objective score means. In addition, the variation in objective and subjective Greenness and Enclosure scores indicates that people are less sensitive to the exact number of perceived Greenery in a scene indicated by tree view, while they are more sensitive to the perception of enclosure (or openness) than the single indicator of sky view. Such divergence between two measurement systems indicates that the underlining mechanism of subjective perception would be quite different from objective formulae. Summing up or recombining view indices of selected visual elements cannot reflect all factors exhaustively with some unobserved factors that can never be captured.

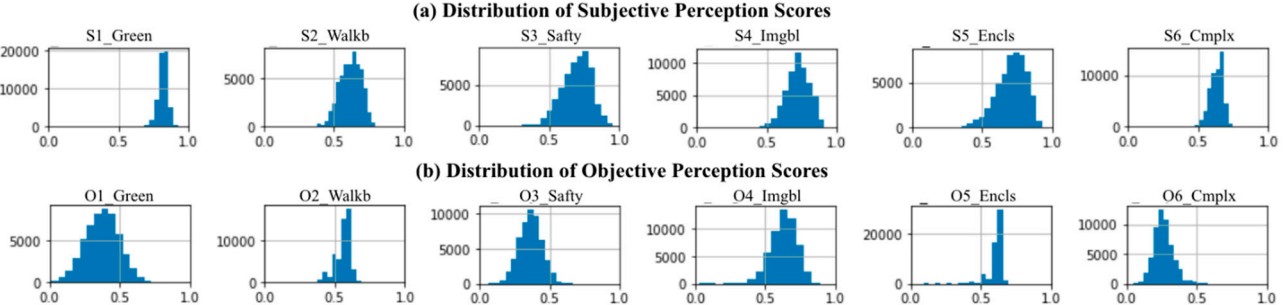

**Figure 9.** Score distribution. (**a**) Histogram of six subjective scores. (**b**) Histogram of six objective scores.

Meanwhile, five SVI samples were randomly selected with the segmentation results and perception scores illustrated by Figure 10. The scores of the six perceptual qualities are shown with radar charts with divisions of ten levels from 0 to 1 from the inside out. Greenness, Complexity, and Safety are found to exhibit larger differences, while the Walkability, Imageability, and Enclosure scores are relatively closer.

### 4.3. Hedonic Price Model Selection

We first tested the overall importance in explaining housing prices for the five groups of attributes (see Table 6), using coefficient of determination ($R^2$) as a criterion. Our study indicated the ranking of importance as follows: location (0.678) > neighborhood (0.556) > subjective streetscape scores (0.322) > structural attributes (0.188) > objective streetscape scores (0.068).

**Table 6.** Model performance with different groups of attributes.

| OLS Diagnosis | Structure Attributes | Location Attributes | Neighborhood Attributes | Subjective Streetscape Score | Objective Streetscape Score |
|---|---|---|---|---|---|
| **Adjusted R2** | 0.188 | 0.678 | 0.556 | 0.322 | 0.068 |
| **Pro (F-statistic)** | 0.000 *** | 0.000 *** | 0.000 *** | 0.000 *** | 0.000 *** |
| **Durbin–Watson** | 1.986 | 1.999 | 1.9835 | 1.998 | 2.003 |

Notes: *p* value * < 0.1, ** *p* < 0.05, *** *p* < 0.01.

All structure, neighborhood, and location attributes were then incorporated into an OSL model and insignificant variables such as the structure of the building, number of living rooms and kitchens, and some submarket dummy for certain districts were abandoned. In addition, continuous variables associated with large VIF values (>5) indicated moderate to high correlations. We removed the less important variable using Gini importance. For example, distance to service variables were removed because they correlated with accessibility measures, while accessibility outperforms distance measures in Gini score. It is also intuitively reasonable that the convenience to access a bundle of services was more important than being located close to a particular service. We formed the baseline model (Model 1) consisting of significant structure, location, and neighborhood attributes, which explained 78.3% of the housing price variances.

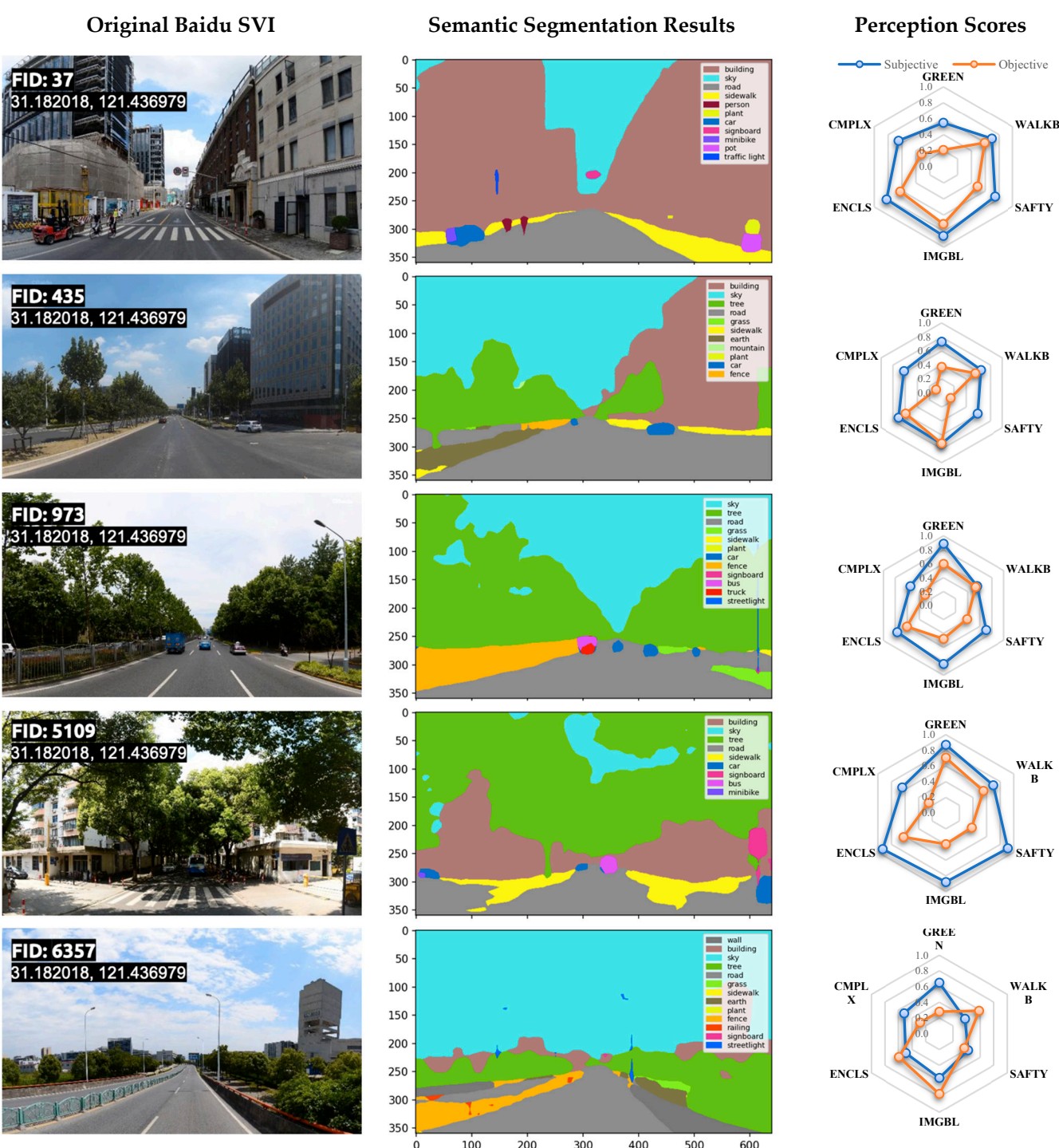

**Figure 10.** Samples of origin SVIs, semantic segmentation results, and the corresponding predicted subjective and derived objective perception scores.

Based on the baseline model, we added five subjective (Model 2) and objective scores (mode 3) except for Complexity score, respectively. Complexity was removed because Pearson's correlation analysis indicated that this quality had strong correlation with at least one or two other qualities, and its VIF indicated serious multicollinearity issues. Table 7 reported the results for the three models. All variables were significant, and most VIFs were smaller than five, indicating no evidence of strong multicollinearity. Appendix Table A1 provides the interpreted economic value of the explanatory variables using averaged coefficients of the three models.

**Table 7.** OLS regression results and diagnosis for the three models.

| Variable | Feature Importance | Model 1 (Base Model) | | | Model 2 (Base + Subjective Scores) | | | Model 3 (Base + Objective Scores) | | |
|---|---|---|---|---|---|---|---|---|---|---|
| | | Coef | std err | VIF | Coef | Std Err | VIF | Coef | Std Err | VIF |
| Constant | / | −0.671 *** | 0.106 | / | −0.691 *** | 0.106 | / | −0.770 *** | 0.106 | / |
| *Structure Attributes* | | | | | | | | | | |
| FLAREA | 0.831 | −0.0002 *** | 0.000 | 5.4 | −0.0002 *** | 0.000 | 5.4 | −0.0002 *** | 0.000 | 5.4 |
| BEDRM | 0.145 | −0.003 *** | 0.001 | 3 | −0.002 *** | 0.001 | 3 | −0.002 *** | 0.001 | 3 |
| BATH | 7.596 | 0.023 *** | 0.001 | 3 | 0.023 *** | 0.001 | 3 | 0.023 *** | 0.001 | 3 |
| CSTRYR | 0.831 | 0.003 *** | 0.000 | 2.3 | 0.003 *** | 0.000 | 2.3 | 0.003 *** | 0.000 | 2.3 |
| ELEVTR | 0.831 | 0.039 *** | 0.001 | 3.5 | 0.040 *** | 0.001 | 3.6 | 0.039 *** | 0.001 | 3.6 |
| HGHT | 0.831 | −0.015 *** | 0.001 | 1.2 | −0.015 *** | 0.001 | 1.2 | −0.015 *** | 0.001 | 1.2 |
| TOWER_SLAB | 0.001 | −0.064 *** | 0.001 | 2 | −0.062 *** | 0.001 | 2 | −0.062 *** | 0.001 | 2 |
| STH_NTH | 0.001 | 0.007 *** | 0.001 | 2.1 | 0.007 *** | 0.001 | 2.1 | 0.007 *** | 0.001 | 2.1 |
| REFNDECOR | 0.534 | 0.023 *** | 0.001 | 4.4 | 0.023 *** | 0.001 | 4.7 | 0.023 *** | 0.001 | 4.5 |
| *Location Attributes* | | | | | | | | | | |
| CTY_FX | 2.017 | −0.167 *** | 0.002 | 2.1 | −0.164 *** | 0.002 | 2.1 | −0.172 *** | 0.002 | 2.1 |
| CTY_HK | 2.136 | 0.018 *** | 0.002 | 1.1 | 0.030 *** | 0.002 | 1.1 | 0.023 *** | 0.002 | 1.1 |
| CTY_HP | 2.017 | 0.057 *** | 0.002 | 1.3 | 0.069 *** | 0.002 | 1.3 | 0.069 *** | 0.002 | 1.3 |
| CTY_JA | 2.017 | 0.065 *** | 0.003 | 1 | 0.073 *** | 0.003 | 1 | 0.075 *** | 0.003 | 1 |
| CTY_JD | 2.017 | −0.060 *** | 0.002 | 1.1 | −0.059 *** | 0.002 | 1.1 | −0.059 *** | 0.002 | 1.1 |
| CTY_JS | 2.017 | −0.170 *** | 0.004 | 1.3 | −0.137 *** | 0.004 | 1.3 | −0.168 *** | 0.004 | 1.4 |
| CTY_PD | 2.017 | 0.027 *** | 0.001 | 1.6 | 0.022 *** | 0.001 | 1.7 | 0.027 *** | 0.001 | 1.7 |
| CTY_PT | 1.407 | −0.021 *** | 0.002 | 1.3 | −0.014 *** | 0.002 | 1.3 | −0.013 *** | 0.002 | 1.3 |
| CTY_QP | 0.831 | −0.050 *** | 0.003 | 1.2 | −0.057 *** | 0.003 | 1.3 | −0.057 *** | 0.003 | 1.3 |
| CTY_SJ | 0.831 | −0.050 *** | 0.002 | 1.2 | −0.046 *** | 0.002 | 1.2 | −0.053 *** | 0.002 | 1.2 |
| CTY_YP | 0.831 | 0.033 *** | 0.002 | 1.1 | 0.041 *** | 0.002 | 1.2 | 0.038 *** | 0.002 | 1.2 |
| CTY_ZB | 0.831 | 0.022 *** | 0.002 | 1.9 | 0.024 *** | 0.002 | 2 | 0.029 *** | 0.002 | 1.9 |
| LND2CTR | 0.534 | −0.109 *** | 0.001 | 1.3 | −0.108 *** | 0.001 | 1.4 | −0.108 *** | 0.001 | 1.4 |
| **Neighborhood Attributes** | | | | | | | | | | |
| LNDENWRK | 0.534 | 0.002 *** | 0.000 | 1.2 | 0.002 *** | 0.000 | 1.2 | 0.002 *** | 0.000 | 1.2 |
| LNDENSRV | 0.534 | 0.003 *** | 0.000 | 1.3 | 0.001 *** | 0.000 | 1.4 | 0.002 *** | 0.000 | 1.3 |
| LNA2MTR | 0.534 | 0.021 *** | 0.000 | 2.3 | 0.021 *** | 0.000 | 2.3 | 0.021 *** | 0.000 | 2.3 |
| LNA2SCH | 0.534 | 0.053 *** | 0.001 | 1.4 | 0.051 *** | 0.001 | 1.4 | 0.052 *** | 0.001 | 1.4 |
| **Subjective Street Scores** | | | | | | | | | | |
| S1_GREEN | 0.534 | / | / | / | −0.327 *** | 0.015 | 2.5 | / | / | / |
| S2_WALKB | 0.475 | / | / | / | −0.189 *** | 0.009 | 4.1 | / | / | / |
| S4_SAFTY | 0.001 | / | / | / | 0.188 *** | 0.010 | 7.7 | / | / | / |
| S4_IMGBL | 0.001 | / | / | / | 0.134 *** | 0.008 | 3.6 | / | / | / |
| S5_ENCLS | 0.001 | / | / | / | −0.040 *** | 0.010 | 8.9 | / | / | / |
| **Objective Street Scores** | | | | | | | | | | |
| O1_GREEN | 0.534 | / | / | / | / | / | / | 0.034 *** | 0.006 | 4.8 |
| O2_WALKB | 0.534 | / | / | / | / | / | / | −0.013 * | 0.007 | 1.4 |
| O3_SAFTY | 0.534 | / | / | / | / | / | / | 0.053 *** | 0.005 | 1.2 |
| O4_IMGBL | 0.534 | / | / | / | / | / | / | −0.074 *** | 0.008 | 4.8 |
| O5_ENCLO | 0.534 | / | / | / | / | / | / | −0.030 *** | 0.011 | 1.6 |
| **Diagnosis** | | | | | | | | | | |
| *Adj. R2* | | 0.783 | | | 0.791 | | | 0.787 | | |
| *Prob (F-statistic)* | | 0 *** | | | 0 *** | | | 0 *** | | |
| *Durbin–Watson* | | 2.009 | | | 2.007 | | | 2.007 | | |
| *No. Observation* | | 40,159 | | | 40,159 | | | 40,159 | | |

Note: ***, **, and * indicate significance level of 1%, 5% and 10%, respectively.

### 4.3.1. Streetscape Perception Attributes

Subjective measures significantly outperformed the objective counterparts in explaining house price: the former explained 32.2% data variance, while the latter only explained 6.8% (see Table 6). Comparing Model 2 and Model 3, first, the impacts of streetscape scores were all significant at the 0.01 confidence interval except for the objectively measured Walkability. Their coefficients were all non-negligible. However, their contributions to the overall goodness-of-fit improvement were minimal, with 0.08 and 0.04 larger R2 values compared to the base Model 1, respectively. Their feature importance

score ranking indicated that besides Greenness, all other subjective measures had stronger explanation power than the objective counterparts (Figure 11a). In addition, subjectively measured Walkability had the largest importance score.

Most importantly, the results indicated coherence as well as large divergence. On the one hand, while three perceptual qualities, i.e., Walkability, Safety, and Enclosure, implied consistent signs in their subjective and objective measures, the other two also showed large divergence in their coefficient magnitudes (Figure 11b). On the other hand, two subjective and objective scores, i.e., Greenness and Imageability, exhibited opposite signs (Figure 11c). With a 10% increase in Greenness score, the subjective measure was correlated with a −3.3% (or −1876 RMB) decrease in house prices while objective counterpart was correlated with a 0.3% increase. In addition, for Imageability score, a 10% increase in the subjective score was correlated with a 1.3% increase while objective measure saw a −0.7% decrease. These demonstrate future areas for further studies.

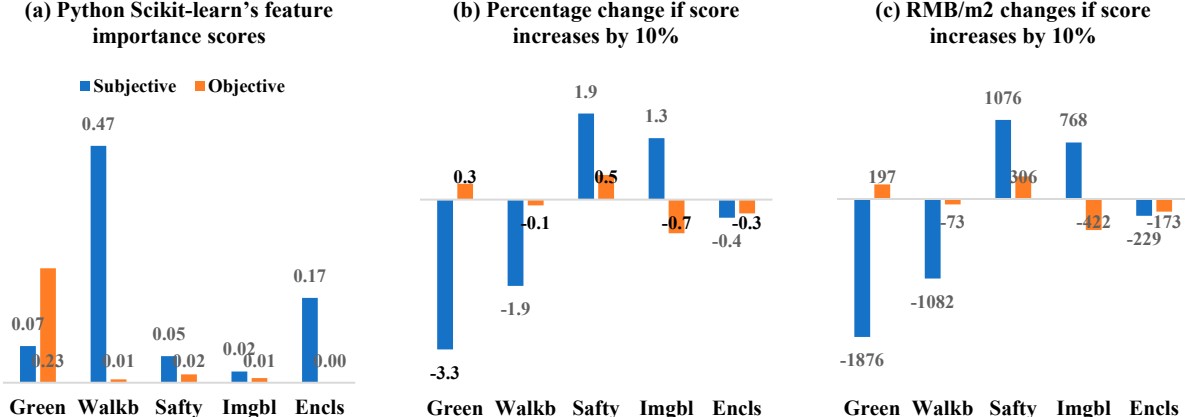

**Figure 11.** Comparing subjective and objective streetscape scores in (**a**) global importance ranking, (**b**) impact on housing price percentage changes and (**c**) per square meter price change if score increases by 10%.

### 4.3.2. Location Attributes

Location attributes were the most dominant, explaining 67.8% of the price variance. First, centrality to city center represents the level of potential services and have implications on living costs such as commuting and education, therefore the distance to city center largely affected sales price. Second, whether located in certain districts captured the fixed spatial effects that are highly correlated the larger-scale neighborhood quality; therefore, its price premium incorporated willingness to pay for being closer to school districts and metro stations from neighborhood attributes. In general, with other variables constant, sales price decreased by approximately 1.1% with a 10% increase in distance to CBD. Given the average distance of 12.6 km and average house price of 57,349 RMB/m², per square meter house price decreased by 4935 RMB if it is 10 km or greater from CBD.

Price premiums for certain submarkets were identified by submarket dummy variables (e.g., the CTY_XX variables indicating property was within the administrative boundary). For example, on average, prices in Jing'an were 6.45% (or 3699 RMB/m²) more expensive than the average, while prices in Fengxian were 13.7%–17.03% (or 7857 to 9767 RMB/m²) cheaper than average.

### 4.3.3. Neighborhood Attributes

The density of working opportunities (lnDenWrk), the density of living services (lnDenLiv), and accessibility to schools (lnA2Schl) and metro stations (lnA2Metro) were all significant as included in neighborhood attributes and they explained 55.6% of the data variance. All neighborhood variables were positively related to house prices, which was consistent with the literature [21,93]. Particularly, a remarkable price premium has been

found in school districts and subway stations: with every five kilometers away from the nearest metro station or excellent high schools recognized by Shanghai government, the house prices dropped by 8.8% and 2.5%, respectively.

### 4.3.4. Structure Attributes

The structural attributes collectively explained 18.8% of the variance. The signs of their coefficients were consistent with the literature. With other variables being constant, apartments with a refined interior design were sold at 2.3% (or 1330 RMB/m²) more. The attributes of south-facing rooms bring the sales price up by 0.7%. In addition, apartments with elevators exhibit a price premium of 3.9%.

## 5. Discussion

### 5.1. The Significance of Streetscape Perceptual Qualities

First, the HPM studies implied that house prices in Shanghai can be significantly influenced by both subjectively assessed and formula-derived objective measures of streetscape perception. Second, using coefficient of determination as a criterion, the ranking of the importance for five attribute groups was as follows: locational > neighborhood > subjective streetscape scores > structural > objective streetscape scores. Specifically, except for Greenness score, all other subjectively assessed qualities (i.e., Walkability, Safety, Imageability, and Enclosure) largely outperformed objective counterparts in explaining house prices variance.

However, the objective measure also has advantages when the perception definition is clear, and its operationalized protocol captures fewer visual elements. In our case, objectively measured Greenness won the subjective counterpart (Figure 11a) and the sign of its coefficient was also consistent with much of the literature [18,19,21,25] that street greenery was positively related to house price. In our study, a 10% increase in the objective Greenness score was related to a 0.3% (or 197 RMB/m²) increase in transaction price, which was smaller in magnitude than a prior study [18], where every one percent increase in the GVI increased housing prices by 71 RMB/m².

Notably, traditional location and neighborhood attributes, such as being in a certain sub-city (district), the accessibility to school, the distance to CBD, and the accessibility to metro stations, were still among the top six determinants of house prices (see Table 7 regarding feature importance). Three structure variables, including decoration, elevator, and number of bathrooms, were also among the top ten. However, perceptual qualities such as subjective Walkability were more important than having a south-facing room, and objective Greenness score and subjective Enclosure score were more important than the floor height in the building, number of bedrooms, and building age.

While increasing studies with deep learning and SVIs focus on the influence of single or multiple visual elements, this study implied the non-negligible impacts of more comprehensively measured human perception on house prices. Particularly, objective Greenness, subjective Safety and Imageability scores indicated a positive relationship to house prices. In other words, a better street environment in these qualities provided non-negligible price premiums to real-estate developers, while the cost of investment and maintenance in public street's streetscape was mainly taken care of by the cities.

More specifically, in practice, developers in Chinese cities, such as Guangzhou, are required to pay fees to cities, with commonly a portion of the cost for constructing surrounding infrastructure, especially in new town projects [75,76], as a practice of local land value capture [77]. Developers also compete to offer good landscape and environment to lure buyers [78]. Increasing attention has been devoted to beautifying residential grounds and landscape designs within the gated community [78,95]. However, the general urban design of these streets is still determined solely by the will of cities (e.g., urban designers and planners). The implicit value of environmental design quality of public streets is seldom incorporated into the valuation of properties [18,25,26,78]. Although it is intuitive for

most developers to believe the value of green spaces and pleasant neighborhood environment, compared to investments in apartment building constructions, inputs into engaging and facilitating beatifying streets are still not sufficient. While the implicit return from investing in streetscapes on improving property values is noneligible [32,36,96], our finding suggests that the urban design process for deciding the streetscape could be more participatory, allowing different stakeholders to contribute to a better street environment [30]. On the other hand, our findings asserted that while real-estate developers have already benefitted from the surrounding street environment, they should have taken more responsibility such as contributing to maintaining the street greenery [21,25]. Hence, municipal government could potentially levy a street environment tax to compensate the public budget invested in designing and maintaining street environments where a property price premium has been enjoyed by developers. The tax amount could be calculated according to the method established in our study if the perception score of a residential unit is within the range.

In addition, the findings also advocate urban planners and real-estate developers to not limit their focus on the micro-level environment such as the tree canopy within residential blocks, but also extend their attention to the public domain: street-level neighborhood environment that pedestrians and residents perceive on the daily basis. Street perception scores can be used as a novel metric for street design and urban design guidelines, and can inform urban renewal strategies [35]. Most importantly, better street perceptual qualities provide improved streetscape aesthetics and appreciation during residents' activities and incentivize home buyers' willingness to pay.

### 5.2. Coherence and Divergence between Subjective and Objective Measures

This study also demonstrated promising areas for future studies which call for more efforts to stress the coherence and divergence of the two measurements. First, the subjective and objective measures of Greenness and Imageability implied opposite signs in affecting house prices: being consistent with the literature [19,25], the objective Greenness was positively related to house prices, while the subjective counterpart exhibited a negative sign. This might be due to subjective Greenness, which captured more factors than simply the tree canopy. Second, the subjective Imageability indicated a positive association while its objective counterpart depicted a negative sign.

In short, simply summing up or recombining visual elements could not comprehensively capture or represent more comprehensively defined perceptual quality. There are underlying mechanisms that relate to the psychological, social-demographical characteristics of street users that cannot be exhaustively incorporated by view indices or recombination of them but were significantly affecting home buyers' willingness to pay. Our results indicated that when choosing between subjective and objective measurement, the familiarity to the perceptual quality's operationalized definition is the driving factor for daily street users to make final decisions. Objective measures might outperform subjective measures when perceptual quality is self-evident and not complicated, such as the Greenness score. For the other four dimensions—Walkability, Safety, Imageability, and Enclosure—whose concepts were not familiar to the average person [37], a subjective framework exhibits better performance over objective counterparts.

### 5.3. The Effectiveness of the Integrated Big Data Framework

Few studies have stressed the economic value of subjectively measured human-scale perception quality considering housing prices at a large scale. Prior studies that focused on streetscape as a determinant were constrained to top-down indicators from GIS and remote sensing imagery datasets, including tree canopy area, green area ratio in land use, and the distance to parks. Although previous studies in this regard focus on human eye-level perception, only minor attention was focused on objective features, such as the view index of tree, sky, and building. Our study provides a comprehensive framework for both subjective and objective measures in six important perceptual qualities and integrated

crowdsourcing and open-source SVIs to establish an automated approach. The framework offers strong generalization capability and can be applied across scales where SVIs are available.

## 6. Conclusions

This study proposed a new approach for urban-scale application to quantify both subjective and objective human-scale streetscape perceptual quality. Built on prior quantitative studies on urban design quality [35,60,97] and emerging applications in deep learning and SVIs in urban scene perceptions [62,82], we integrated and extended existing frameworks to (1) effectively collect and evaluate both subjectively and objectively measured perceptions; (2) investigate the coherence and divergence in ML-predicted subjective scores and formula-derived objective scores; and (3) compare the effects on house prices with the two perception measurements taking Shanghai as a case study.

Particularly, we investigated the divergence and coherence between subjective and objective measures for six perceptual qualities, i.e., Greenness, Walkability, Safety, Imageability, Enclosure, and Complexity. We quantified their associations with housing price variations in Shanghai. First, regarding the collective explanatory power within each attribute group, subjective scores explained more variance over structural attributes and objective scores. Second, the percentage increase in sales price attributable to perceived street quality is significant for both subjective and objective measurements. Except for Greenness score, all other subjectively measured qualities outperformed objective counterparts. Particularly, objective Greenness, subjective Safety, and Imageability scores positively affected house prices in Shanghai. Objective Greenness was more important than the structure attributes of floor height, number of bedrooms, and building age, which were conventionally conceived as important in HPM. This is the first study comprehensively expanding HPM with both subjectively and objectively measured streetscape qualities. We suggested that city authorities could levy a street environment tax to compensate the public budget invested in street environments where developers secured benefits from a price premium.

Second, this study also sheds light on promising areas for future studies which call for the coherence and divergence of the two measurements to be further stressed. Specifically, for Greenness and Imageability scores, the subjective and objective measures implied opposite signs in affecting house prices. On the one hand, there might be mechanisms related to the psychological, social-demographical characteristics of street users that have not been fully incorporated by view indices or recombination of them but were significantly affecting home buyers' willingness to pay. On the other hand, when choosing between subjective and objective measurement, final decision could be made based how straightforward the perceptual quality's operationalized definition is to daily street users. Objective measures might outperform subjective measures when perceptual quality is self-evident and not complicated, for example the Greenness. For perceptual qualities that were not familiar to the average, a subjective framework exhibits better performance. The strong generalization capability of this study also called for expanding the measurement of streetscape by incorporating street thermal comfort given that the unique perspective of SVIs can better reflect the vertical dimension of urban geometry.

### Limitations

There are several limitations: (1) failing to indicate causal inference with a more comprehensive panel dataset; (2) ignoring the endogenous effect on green space and housing price, (3) data quality and the prediction accuracy of subjective perceptions could be improved; and (4) the weak interpretability of perception scores.

First, we must acknowledge that the intention was not to make any causal statements. Instead, this study aimed to use correlation to justify the effort and value of incorporating extra micro-scale urban perception data and provide references for selecting measurement methods. Association does not imply causation, as there could be a reverse causal

relationship, or a confounding factor with housing prices and street environments that was omitted. Moreover, as our reviewer pointed out, research findings can only be used as a reference by policy makers when there is a clear causal pathway. Whether simply improving certain indicators can consequently improve real-estate market values still needs to be further demonstrated, especially when it comes to recommendations to decision makers. We should be cautious about any biased estimations in cross-sectional data that overestimate/underestimate the monetized value of street environments, which might result in wasting/lacking public investments.

This also points to an important area of consideration—future studies should carefully design the model procedure to validate that the findings are stable. The causality relationships between housing price and the many variables can provide much more convincing policy recommendations to decision makers as well as more profound empirical results that will largely enrich the literature. Future studies can plan for more serious panel or pseudo panel data to investigate the causal relationship.

Second, thanks to our reviewers, even before taking policy effect into consideration, the green space inside communities is indeed developed by property developers themselves. Considering strategic behavior, greenery will be endogenous in the regression of housing price on greenery, and the research question asked in this work could be a false proposition.

Third, data source and prediction accuracy could be further improved. On the one hand, housing data acquisition was limited by the real-estate website which indispensably contains missing or biased data, which might lead to bias in model estimation. Although the street-level images acquired help us understand the quality of public streets, the impacts of private streets of inner blocks remained unknown due to the lack of SVI data. Moreover, current subjective scores were collected from a specific and small study group—designers. Inputs from potential homebuyers will be more desirable and would likely shed light on more relevant user preferences. Future studies can work on homebuyers' street scene preferences by randomly selecting group of people who visit real-estate offices.

On the other hand, the prediction of subjective perceptual scores has much room for improvement. We intentionally took a rule-based approach [55] using only high-level features (i.e., streetscape view indexes) for prediction in order to align with urban design-oriented measures [37] and ensure interpretability for designers. However, we must acknowledge that incorporating low-level features can complement high-level features to significantly improve prediction accuracy [61,87,88]. Meanwhile, collecting a large training dataset with inputs of more raters can also improve the results [1,2].

Fourth, the street view scores could be difficult to interpret. Future studies can focus on converting these scores into more actionable urban design guidelines and interventions that facilitate better streets. Moreover, since the method seems highly scalable and applicable, applying it to other cities to discover common or divergent impacts of street perception on property values would be highly desirable. Additionally, the proposed approach has demonstrated the capability to assess the economic value of thermal comfort and heat-related design considerations in streets given its capacity to proxy streetscape across scales. More factors that directly or indirectly relate to streetscape and can affect housing prices could be captured in future studies.

**Author Contributions:** Conceptualization, W.Q., X.X. and Z.Z.; methodology, W.Q., W.L. and X.L. (Xiaojiang Li); software, W.Q., W.L and X.L. (Xiaojiang Li); validation, W.Q. and X.X.; formal analysis, W.Q.; investigation, W.Q.; resources, W.Q., X.L. (Xun Liu) and D.L.; data curation, W.Q. W.L. and X.L. (Xun Liu); writing—original draft preparation, W.Q., X.X. and Z.Z.; writing—review and editing, W.Q., W.L., Z.Z. and X.X.; visualization, W.Q. and X.L. (Xun Liu); supervision, W.Q.; project administration, W.Q. and X.X.; funding acquisition, W.Q. All authors have read and agreed to the published version of the manuscript.

**Funding:** This research was supported by the Kermit C. & Janice I. Parsons Scholarship (2019) and the Portman Family Graduate Student Award (2021) from the Department of City and Regional Planning, Cornell University.

**Institutional Review Board Statement:** Ethical review and approval were waived for this study due to the analyzed datasets are properly anonymized, no participant can be identified.

**Informed Consent Statement:** Written informed consent was waived as the analyzed dataset was properly anonymized, so no participant can be identified.

**Data Availability Statement:** The data presented in this study are available on request from the corresponding author.

**Acknowledgments:** We would like to thank the participants who helped to rank SVI perception qualities in the Digital Future Workshop (2020) as well as the sponsor and organizer of the workshop—Prof. Philip YUAN and Tongji University. Additionally, this research was supported by the Kermit C. & Janice I. Parsons Scholarship (2019) and the Portman Family Graduate Student Award (2021) from the Department of City and Regional Planning, Cornell University.

**Conflicts of Interest:** The authors declare no conflict of interest.

## Appendix A

**Table A1.** Interpretation of regression coefficients (converted to RMB/m$^2$).

| | Model1 | Model2 | Model3 | Average | Delta X | Mean X |
|---|---|---|---|---|---|---|
| **Structure Attributes** | | | | | | |
| FLAREA | −11.5 | −11.5 | −11.5 | −11.5 | 1 unit change | 85m$^2$ |
| BDRM | −154.8 | −114.7 | −137.6 | −135.7 | | 2.1 |
| BATH | 1342.0 | 1307.6 | 1307.6 | 1319.0 | | 1.2 |
| CSTRYR | 160.6 | 166.3 | 160.6 | 162.5 | | 1998 |
| ELEVTR | 2248.1 | 2265.3 | 2225.1 | 2246.2 | Y/N (1/0) | 0.4 |
| HGHT | −843.0 | −860.2 | −854.5 | −852.6 | Y/N (1/0) | 0.4 |
| TOWER_SLAB | −3641.7 | −3549.9 | −3532.7 | −3574.8 | Y/N (1/0) | 0.09 |
| STH_NTH | 412.9 | 372.8 | 401.4 | 395.7 | Y/N (1/0) | 0.8 |
| DÉCOR | 1330.5 | 1301.8 | 1313.3 | 1315.2 | Y/N (1/0) | 0.52 |
| **Location Attributes** | | | | | | |
| CNTY_FX | −9560.1 | −9422.4 | −9864.0 | −9615.5 | Y/N (1/0) | 0.025 |
| CNTY_HK | 1026.5 | 1726.2 | 1290.4 | 1347.7 | Y/N (1/0) | 0.038 |
| CNTY_HP | 3280.4 | 3974.3 | 3934.1 | 3729.6 | Y/N (1/0) | 0.032 |
| CNTY_JA | 3699.0 | 4197.9 | 4278.2 | 4058.4 | Y/N (1/0) | 0.024 |
| CNTY_JD | −3418.0 | −3395.1 | −3400.8 | −3404.6 | Y/N (1/0) | 0.041 |
| CNTY_JS | −9766.5 | −7856.8 | −9611.7 | −9078.3 | Y/N (1/0) | 0.055 |
| CNTY_PD | 1525.5 | 1278.9 | 1548.4 | 1450.9 | Y/N (1/0) | 0.234 |
| CNTY_PT | −1175.7 | −820.1 | −751.3 | −915.7 | Y/N (1/0) | 0.073 |
| CNTY_QP | −2873.2 | −3286.1 | −3268.9 | −3142.7 | Y/N (1/0) | 0.017 |
| CNTY_SJ | −2878.9 | −2615.1 | −3016.6 | −2836.9 | Y/N (1/0) | 0.055 |
| CNTY_YP | 1892.5 | 2334.1 | 2202.2 | 2142.9 | Y/N (1/0) | 0.077 |
| CNTY_ZB | 1250.2 | 1382.1 | 1645.9 | 1426.1 | Y/N (1/0) | 0.045 |
| lnD2Ctr | −622.8 | −617.1 | −621.1 | −620.3 | 10% change | 12.62 km |
| **Neighborhood Attributes** | | | | | | |
| LN(DENWRK) | 11.5 | 11.5 | 10.3 | 11.1 | 10% change | 9500/km$^2$ |
| LN(DENSRV) | 14.3 | 6.9 | 9.7 | 10.3 | 10% change | 115/km$^2$ |
| LN(A2MTR) | 122.7 | 119.3 | 122.7 | 121.6 | 10% change | 5.7 |
| LN(A2SCH) | 306.2 | 293.6 | 300.5 | 300.1 | 10% change | 7 |
| **Subjective Street Perception** | | | | | | |
| S1_GREEN | / | −1876.5 | / | −1876.5 | 0.1 score change | 0.8 |
| S2_WALKB | / | −1081.6 | / | −1081.6 | 0.1 score change | 0.6 |

| | | | | | | |
|---|---|---|---|---|---|---|
| S4_SAFTY | / | 1075.9 | / | 1075.9 | 0.1 score change | 0.7 |
| S4_IMGBL | / | 768.5 | / | 768.5 | 0.1 score change | 0.7 |
| S5_ENCLS | / | −228.8 | / | −228.8 | 0.1 score change | 0.7 |
| **Objective Street Scores** | | | | | | |
| O1_GREEN | / | / | 197.3 | 197.3 | 0.1 score change | 0.4 |
| O2_WALKB | / | / | −73.4 | −73.4 | 0.1 score change | 0.6 |
| O3_SAFTY | / | / | 306.2 | 306.2 | 0.1 score change | 0.4 |
| O4_IMGBL | / | / | −422.1 | −422.1 | 0.1 score change | 0.6 |
| O5_ENCLO | / | / | −173.2 | −173.2 | 0.1 score change | 0.6 |
| **Y: Average Price** | 57,349 RMB/m² | | | | | |

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
