# Peer review of "Associations between Street-View Perceptions and Housing Prices: Subjective vs. Objective Measures Using Computer Vision and Machine Learning Techniques"

_remotesensing, doi:10.3390/rs14040891_

Round 1

Reviewer 1 Report

This work presents a number of subjective and objective measurements of the built environment and builds connections between them and real estates values. The paper is easy to follow and generally technically sound. I believe this paper has some merits worth sharing regarding conceptual and empirical aspects, e.g., the comparison of subjective and objective measurements of the same concepts. However, due to the following concerns, I cannot recommend it for publication in its current form:

- It may not be a good idea to use the object ratio (the output of the image segmentation) as a feature to predict the proposed subjective/objective metrics. The output of the segmentation is a limited summary of the information about the scene depicted in the image. In fact, the performance of ML models suggests that these methods fail to model the relationship between object ratios and the proposed dimensionality - most with R2 below 0.5.

- In this work, the relationship between the proposed indicators, such as greenery, and housing prices could be reverse causation. The Chinese real estate market requires a certain green space ratio for a different level of residential development, which means by default the higher the housing price of a community, the more greenery spaces inside/nearby should be. As a fundamental limitation of this work, however, the authors haven't considered this important policy's exogenous effect on greenery space when asking the research question and making a conclusion.

- Even before taking policy effect into consideration, the greenery space inside communities is indeed developed by property developers themselves. Considering strategic behavior, greenery will be endogenous in the regression of housing price on greenery, and the research question asked in this work could be a false proposition. 

- Association does not imply causation. Research findings can only be used as a reference by policymakers when there is a clear causal pathway. The proposed indicators may have some relationship with real estate values. However, whether simply improving certain indicators can consequently improve real estate market values still needs to be further demonstrated, especially when it comes to recommendations to decision-makers.

- several recent and very relevant works are missing in the literature review, for instance: 
1. Wang, Ruoyu, et al. "The distribution of greenspace quantity and quality and their association with neighbourhood socioeconomic conditions in Guangzhou, China: A new approach using deep learning method and street view images." Sustainable Cities and Society 66 (2021): 102664.

2. Kang, Yuhao, et al. "Human settlement value assessment from a place perspective: Considering human dynamics and perceptions in house price modeling." Cities 118 (2021): 103333.

Reviewer 2 Report

Very interesting study. I agree with the methodological assumptions made. They are logically justified and relate to other studies. It is very difficult to find correlation of subjective feelings and impressions with objective calculations. The authors have managed to identify such a correlation in part. I have not read such a good study for a long time. Conclusions result from the conducted research. To make the article even more scientific, I propose to underline the hypothesis that the authors intended to prove. 

Author Response

Please see the attachment, thank you

Round 2

Reviewer 1 Report

The authors have adequately addressed my concerns.